# New data on *Beurlenia araripensis* Martins-Neto & Mezzalira, 1991, a lacustrine shrimp from Crato Formation, and its morphological variations based on the shape and the number of rostral spines

**Olga Alcântara Barros**[1]*, **Maria Somália Sales Viana**[2], **Bartolomeu Cruz Viana**[3], **João Hermínio da Silva**[4], **Alexandre Rocha Paschoal**[5], **Paulo Victor de Oliveira**[6]

**1** Pós-graduação em Geologia, Departamento de Geologia, Universidade Federal do Ceará, Campus do Pici, Fortaleza, Ceará, Brazil, **2** Laboratório de Paleontologia, Museu Dom José, Universidade Estadual Vale do Acaraú, Sobral, Ceará, Brazil, **3** Departamento de Física, Universidade Federal do Piauí, Teresina, Piauí, Brazil, **4** Centro de Ciências e Tecnologia, Universidade Federal do Cariri, Juazeiro do Norte, Ceará, Brazil, **5** Departamento de Física, Universidade Federal do Ceará, Fortaleza, Ceará, Brazil, **6** Laboratório de Paleontologia de Picos, Universidade Federal do Piauí, Campus Senador Helvídio Nunes de Barros, Picos, Piauí, Brazil

* olga.a.barros@gmail.com

**Data Availability Statement:** All relevant data are available from (https://figshare.com/articles/

## Abstract

Fossil freshwater carideans are very rare worldwide. Here, we present new taxonomic remarks about *Beurlenia araripensis* from the Early Cretaceous laminated limestones of the Crato Formation, Araripe Basin, northeastern Brazil. We analyzed five fossil samples, testing the morphological variations such as, rostrum with 5 to 14 supra-rostral spines and 2 to 3 sub-rostral spines, which appears as serrate for Caridea. This variation demonstrates a morphologic plasticity also seen in extant species of the group, such as those of the genera *Macrobrachium* and *Palaemon*.

## Introduction

Among the crustacean fossil fauna worldwide, freshwater decapods are rare, especially in the Mesozoic [1]. In Brazil, Palaemonidae crustaceans are known from the Cenozoic outcrops, with records in the Tremembé Formation (Oligocene of southeast Brazil), and from the Early Cretaceous deposits of the Marizal Formation (northeast Brazil) [2].

The Romualdo and Crato formations of the Araripe Basin show the largest number of fossil shrimp species described for the Cretaceous, including carideans (*Beurlenia araripensis* Martins-Neto & Mezzalira, 1991, and *Kellnerius jamacaruensis* Santana, Pinheiro, Silva & Saraiva, 2013), sergestoids (*Paleomattea deliciosa* Maisey & Carvalho, 1995), luciferids (*Sume marcosi* Saraiva, Pinheiro & Santana, 2018), and penaeoids (*Araripenaeus timidus* Pinheiro, Saraiva & Santana, 2014, *Priorhyncha feitosai* Alencar, Pinheiro, Saraiva, Oliveira & Santana, 2018, and *Cretainermis pernambucensis* Prado, Calado & Barreto, 2019). For intermediate levels of the

dataset/Data_Availability_Statement_/13951607) and the figures used in the research (https://figshare.com/articles/figure/New_data_on_Beurlenia_araripensis_Martins-Neto_Mezzalira_1991/13623023).

**Funding:** The OAB is thankful to the Brazilian Federal Agency for Support and Evaluation of Graduate Education - CAPES (Coordenação de Aperfeiçoamento de Pessoal de Nível Superior), http://capes.gov.br, for the financial support for this research. The funders had no role in study design, data collection and analysis, decision to publish, or preparation of the manuscript.

**Competing interests:** NO authors have competing interests.

Santana Group, the presence of Caridea was also reported for the Ipubi Formation, but the specimens are poorly preserved laterally, hampering a more detailed classification [3].

*Beurlenia araripensis* was described for the first time as a caridean in the family Palaemonidae Rafinesque, 1815. Its description was made considering morphological characters found in a single, almost complete specimen [4]. Years late, with further specimens, additional features were described, especially in the rostral region [5], and the inclusion of *Beurlenia* in Palaemonidae was questioned because no evidence was found of articulated spines or setae on the telson of the new specimens. This, as well as the presence of biflagellate instead of triflagellate antennules and the absence of lateral spines at the tip of the telson further questioned the attribution of *Beurlenia* to palaemonids, thus some authors [5,6] transferred *B. araripensis* to a *incertae sedis* family within Caridea.

Through a new sample of *Beurlenia araripensis*, Saraiva *et al.* [7] reported new inferences about its taxonomy. For example, the carapace previously described as smooth [4] presented two evident spines that, together with articulated spines found in the posterior extremity of the telson, confirmed its inclusion in Palaemonidae. Besides, they also questioned the variation in the number of the rostrum spines (12–14 supra rostral spines) [7], indicating probable morphologic plasticity as occurs for extant species of *Macrobrachium* Bate, 1868 [7].

In this work, we present new taxonomic remarks regarding *Beurlenia araripensis* and observed morphological variations in the 5–14 rostral spines. These have a serrate appearance, confirming the inclusion of *Beurlenia araripensis* in Palaemonidae Rafinesque, 1815.

## Geological setting

The Araripe Basin is one of the most important fossiliferous areas from the Cretaceous Period in the world. The area is known for the excellent preservation, diversity, and quantity of fossils, specifically in the Crato and Romualdo formations. This basin is located in inland northeastern Brazil in the states of Ceará, Pernambuco, and Piauí [8]. This basin is particularly rich in shrimp fossils, with several species preserved in excellent conditions (Fig 1).

The Araripe Basin is subdivided stratigraphically into pre-rift and post-rift (I and II) sequences formed mainly by fluvial and lacustrine strata, including the rich fossiliferous deposit of the Santana Group [8], which comprises the Rio da Batateira, Crato, Ipubi, Romualdo, and Arajara Formations (Aptian-Albian in age) [9].

The three main units with fossils from the Santana Group are the Crato, Ipubi, and Romualdo formations (Fig 1). The Crato Formation is composed by a series of laminated limestones, silts, and silty clays; the fossils of which have been dated as Aptian in age. The Ipubi Formation is composed of evaporites with some intercalated shales, forming a sedimentary section with a maximum thickness of 30 m. The Romualdo Formation includes a heterogeneous sequence of bituminous shales, marls, sandstone and carbonate sediments [10].

The Crato Formation is a *Lagerstätte*, with exceptionally well preserved fossils. The gonorynchiform fish *Dastilbe crandalli* Jordan, 1910 is extremely common [11], but it is also possible to find frogs [12], lizards [13], turtles [14], pterosaurs [15], plants [16], insects [17], and crustaceans [4] in this formation.

## Material and methods

### Paleontological ethics statements

The specimens are deposited in the Laboratory of Paleontology at Universidade Federal do Ceará (UFC, Brazil), in the Laboratory of Paleontology at Universidade Regional do Cariri (URCA, Brazil), in the paleontological collection of the Plácido Cidade Nuvens Museum (MPSC, Brazil), and in the collection of the Geological Department at Universidade Federal do

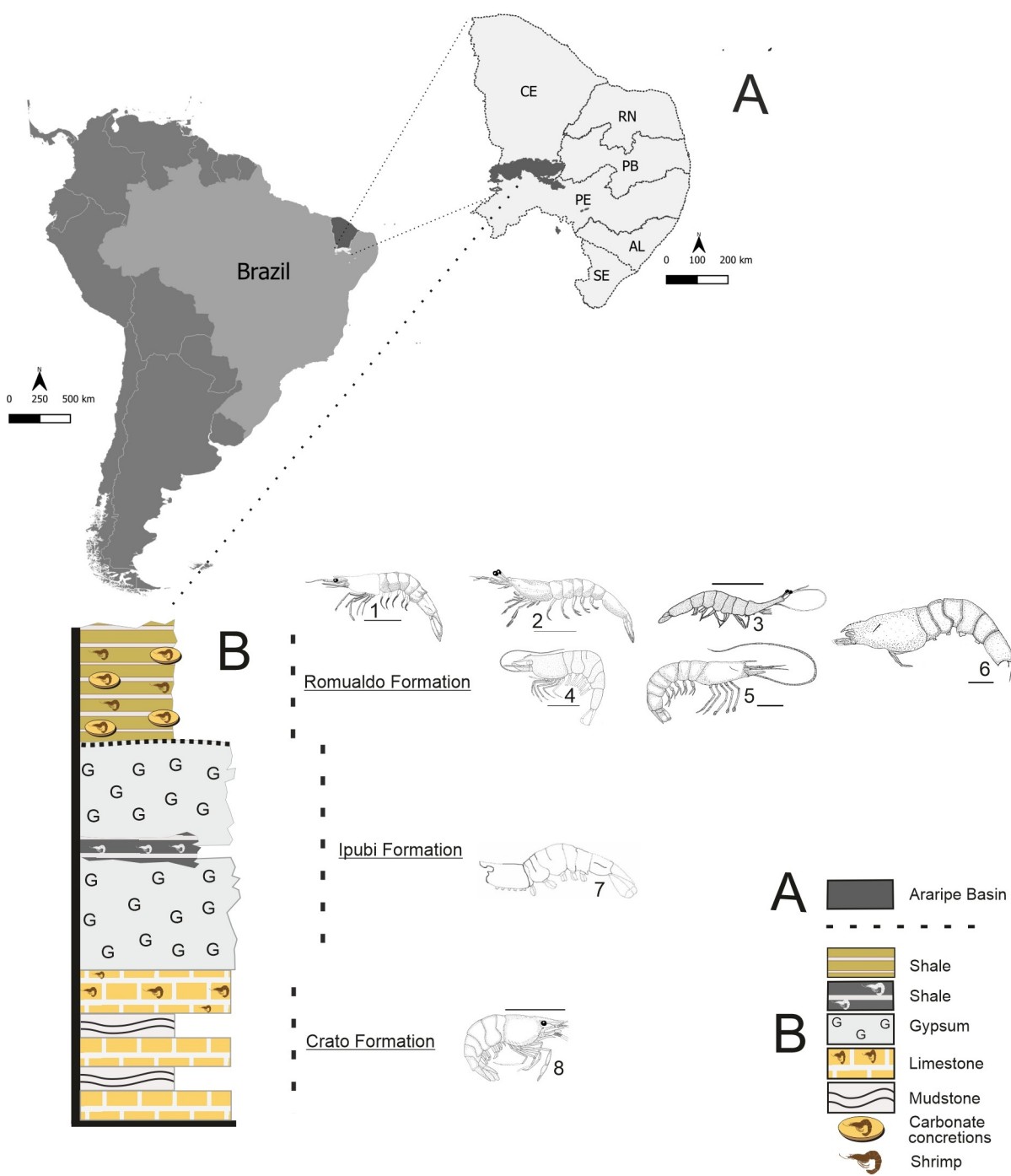

**Fig 1. South America map with Brazil highlighted.** (A) Northeastern Brazil with location of the Araripe Basin (south of the Ceará State). (B) Shrimps that are found in Santana Group: 1. *Araripenaeus timidus*, 2. *Paleomattea deliciosa*, 3. *Sume marcosi*, 4. *Kellnerius jamacaruensis*, 5. *Cretainermis pernambucensis*, 6. *Priorhyncha feitosai*, 7. indeterminate Caridea, 8. *Beurlenia araripensis*, drawn by Olga A. Barros with base in the descriptions: [3,7,9–13].

Rio de Janeiro (UFRJ, Brazil). All the studied samples were taken from the above institutions with the necessary permits, complying with every applicable regulation. All the images analyzed in this work were obtained by the authors with proper permissions from the institutions listed above.

## Analysis of the samples and sampling design

We analyzed the samples using a stereomicroscope. The images were taken on an Olympus C011 trinocular microscope, coupled with a CCD camera, and captured by the Infinity Capture software. The samples were drawn and measured under a stereomicroscope with a Camera Lucida and improved with a drawing table, Parblo A610 –Graphic Tablet. The geographical location of the Araripe Basin was produced using the software QGIS Geographic Information System (version 3.12 –QGIS.org) considering the coordinate system Datum—SIRGAS 2000 from Instituto Brasileiro de Geografia e Estatística (IBGE, Brazil) and Companhia de Pesquisa de Recursos Minerais (CPRM, Brazil). The stratigraphy of the Santana group was drawn by the authors, based on Valença *et al.* [9].

## Results

We analyzed thirteen samples, nine from UFRJ, two from UFC, and two from URCA, but only two samples from UFRJ (171/CR and 243/CR), two from URCA (1673/LPU and 2108/MPSC), and one from UFC (2466/CRT) contributed to the data shown in this work because the other specimens were mostly fragmented or incomplete, without relevant anatomical details.

Systematic Palaeontology

Crustacea Brünnich, 1772

Decapoda Latreille, 1802

Caridea Dana, 1852

Palaemonidae Rafinesque,1815

*Beurlenia* Martins-Neto & Mezzalira,1991

*Beurlenia araripensis* Martins-Neto & Mezzalira,1991

**Stratigraphic Unit:** Crato Formation, Santana Group, Araripe Basin, Ceará, Brazil. Valença *et al.* [9].

**Type Species:** *Beurlenia araripensis* Martins-Neto & Mezzalira, 1991, by original designation.

## Emended diagnosis

Palaemonid with antennae longer than the body; antennules with two flagellae. Rostrum short or medium-sized, with 5 to 14 supra-rostral spines, small and close together with serrate appearance; 2 to 3 sub-rostral spines; rostrum with laminar scaphocerite; third maxilliped with a brush of hair (bristles) and a calcified and hard protuberance that looks like a tubercle; antennal and branchiostegal spines present in the carapace; pleon is six-segmented and smooth without spines; pleurae of the first and third somites are somewhat rounded, while the second somite is strongly rounded, overlapping the first and third ones; fourth and fifth somites have a strongly acute pleura; first two pairs of pereiopods represented by chelipeds (P1-P2), in which the second pereiopod (P2) is enlarged and larger than the first (P1) and contains several tubercles; pereiopods posterior to second pair (P3-P5) are long and narrow; one spine on the merus and several tubercles on the merus, carpus and propodus with elongate dactylus without chelae; multi-segmented pleopods; telson with at least one pair of articulated spines on distal extremity and one articulated spine in the posterior extremity of the telson; uropod (exopod) slightly longer than the telson; exopodite with diaeresis.

## Description of 2466/CRT (UFC) (Figs 2–4)

**Description.** Medium-sized caridean with a well-preserved exoskeleton, total length 46 mm, being one of the largest species found; 4 rostral spines visible, and the last fragmented

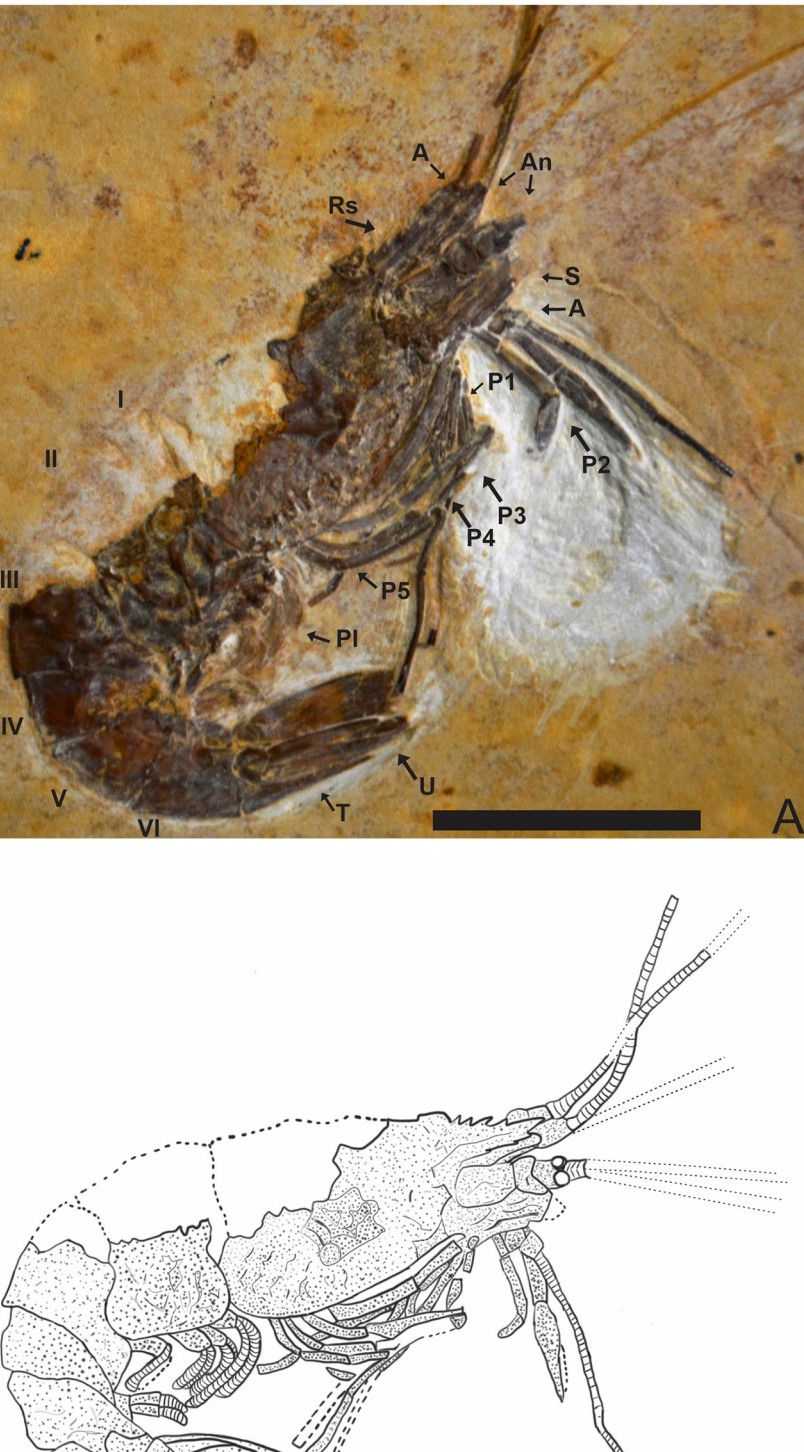

**Fig 2. Lateral view of the specimen 2466/CRT (UFC).** (A) Detail of the chelae on the second pereiopods. The other P3-P5 pereiopods without chelae with elongate dactylus. Pleopods are visible and are multi-segmented. (B) Reconstitution, dotted lines represent appendages not preserved but with impressions on the matrix. Rs: Rostrum; A: Antenna; An: Antennule; S: Scaphocerite; Pl: Pleopods; P1-P5: Pereiopods; I-VI: Pleon; T: Telson; U: Uropods. Scale bar: 10 mm. Photo and drawing by Olga A. Barros.

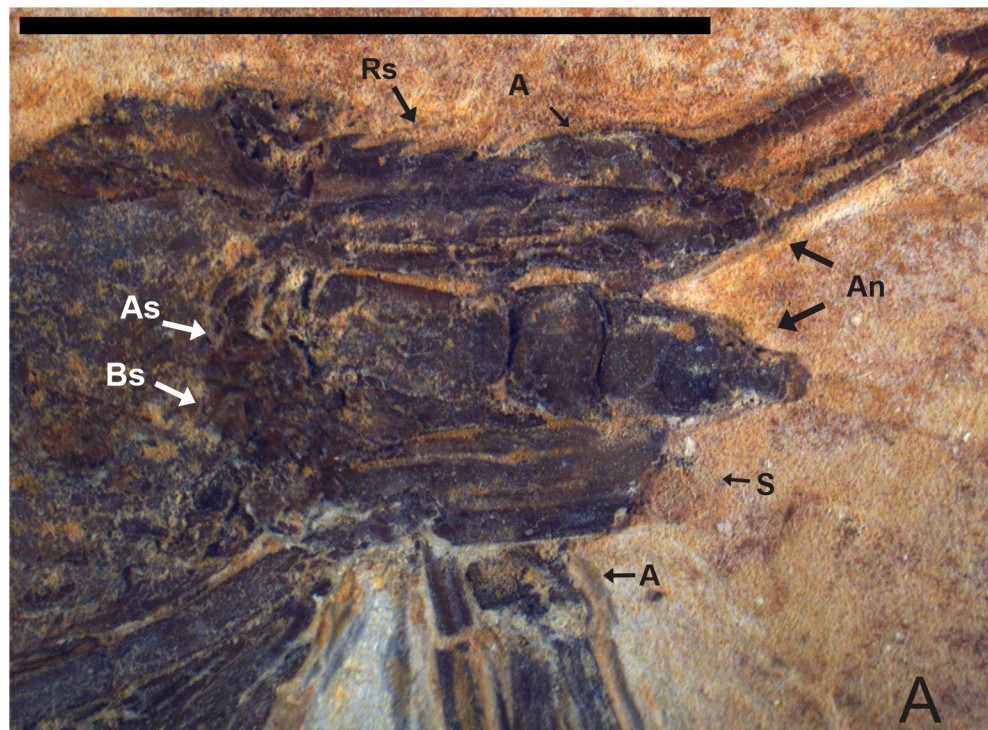

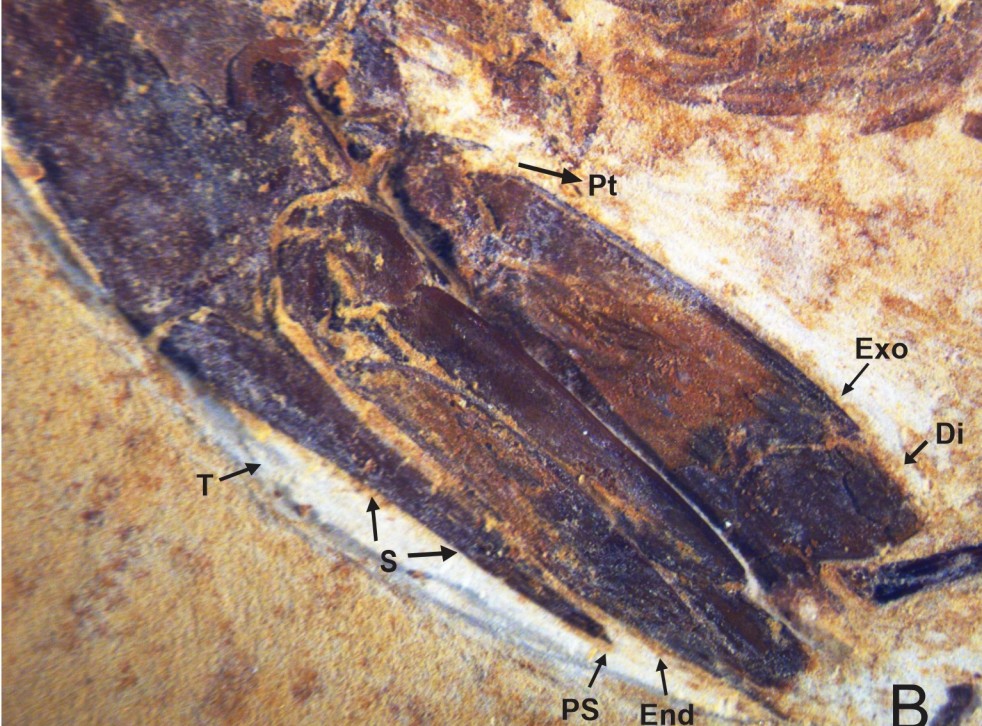

**Fig 3. 2466/CRT (UFC).** (A) Antennal and branchiostegal spines are visible. (B) Detail of the uropods and telson with posterior spines, exopod is slightly longer than the telson. Rs: Rostrum; A: Antenna; An: Antennule; S: Scaphocerite; T: Telson; S: Spines; Ps: Posterior spines; End: Endopodite; Exo: Exopodite; Di: Diaresi; Pt: Protopodite. Scale bar: 10 mm. Photo: Olga A. Barros.

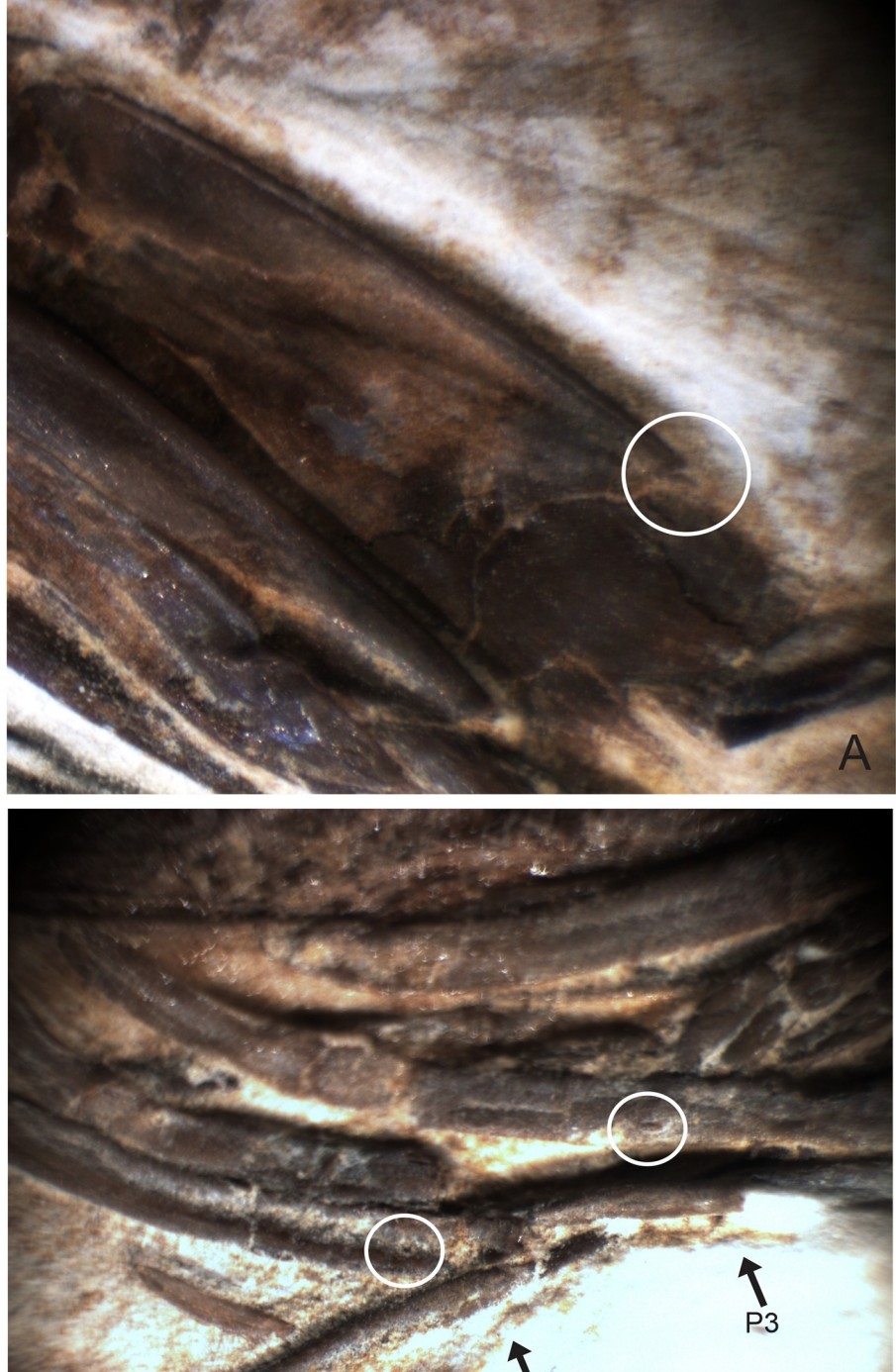

**Fig 4. 2466/CRT (UFC).** (A) Detail of the uropod. Exopodite with diaresis has crossed by a thin longitudinal carina running parallel to the outside lateral margin and ending with a small spine in highlighted. The endopodite without ornamentation. (B) Detail of the Pereiopods with spine.

part at the end of the rostral tip seems to have at 5 spines and 3 sub-rostral spines with serrated appearance; eyestalks not preserved; antennae and antennules very long and incompletely preserved, with unknown total length because of fragmentary nature of specimen; carapace and pleonal pleurae laterally compressed; antennal and branchiostegal spines clearly visible on the carapace; laminar scaphocerite present; pleon six-segmented, all somites smooth without spines, pleura of second somite somewhat rounded, overlapping first and third. Third and fourth somites have strongly acute pleura; pereiopods well preserved, but it is difficult to understand their segmentation; first two pereiopods chelate, but the second (P2) is larger than the first (P1); the subsequent pereiopods (P3-P5) lack chelae and have one spine evident in P3 and P5; pleopods visible and multi-segmented; uropods and telson preserved; telson with at least one pair of articulated spines on distal extremity and one spine in the posterior extremity; uropod slightly longer than the telson; protopodite subrectangular in outline and supports the exopodite. Exopodite, with rounded diaresis, is crossed by a thin longitudinal carina, extending parallel to the outside lateral margin and ending with a small spine. Endopodite without ornamentation.

### Description of 1673/LPU (URCA) (Figs 5–7)

**Description.** The sample has a 46.01 mm total length (rostrum to telson), being one of the largest speciemens found, with a well-preserved exoskeleton; antennae and antennules not well preserved, but it is possible to see their impression on the matrix; rostrum with 12 suprarostral spines with serrate appearance, two sub-rostral spines clearly visible; laminar scaphocerite present; third maxilliped with a brush of hairs (bristles) and a calcified and hard protuberance with a well evident tubercle; carapace laterally compressed, 10.65 mm in total length; antennal and branchiostegal spines clearly visible. Smooth pleon without spines; pleura of the second somite strongly rounded, overlapping the first and third, strongly acute fourth and fifth somites; first two pereiopods chelate, but the second (P2) is larger than the first (P1). First pereiopod (P1) preserved behind the other pereiopods (P2-P4), but it is possible to see the dactylus. Second pair of pereiopods (P2) has several tubercles; first propodus about 4.44 mm long and dactylus 4.26 mm long, second pereiopod of P2 with 3.98 mm long propodus and 4.48 mm long dactylus, both with relatively large sizes when compared with other samples already described; subsequent pereiopods (P3-P5) have one spine on merus and several tubercles on the merus and carpus with elongated dactylus without chelae; pleopods visible and multisegmented; telson and uropod (exopod and endopod) are faintly preserved.

### Description of 2108/MPSC (URCA) (Figs 8 and 9)

**Description.** Specimen with well-preserved exoskeleton; carapace sub-rectangular in lateral view, with 11.5 mm overall length; long antennae and antennules, but total length is unknown because of the fragmentary nature of the specimen; rostrum with 14 supra-rostral spines, small and close together with serrate appearance, and three sub-rostral spines; laminar scaphocerite present; third maxilliped distinctly leg-like in appearance, endopodite with the first podomere represented by fused ischium and merus, second by the carpus, and third by fused propodus and dactylus. Exopodite is smaller than ischium and merus, slender, and unjointed; carapace with clearly visible antennal and branchiostegal spines; smooth pleon without spines, pleura of second somite strongly rounded, overlapping first and third. Fourth and fifth somites with a strongly acute pleura; first two pereiopods chelate, with second pair (P2) larger than the first (P1); pereiopods P3-P5 lacking chelae, long, and narrow, only third pereiopod (P3) has a preserved spine on merus; pleopods multisegmented, telson with at least one pair of articulated spines on distal extremity and one spine in posterior extremity; uropod

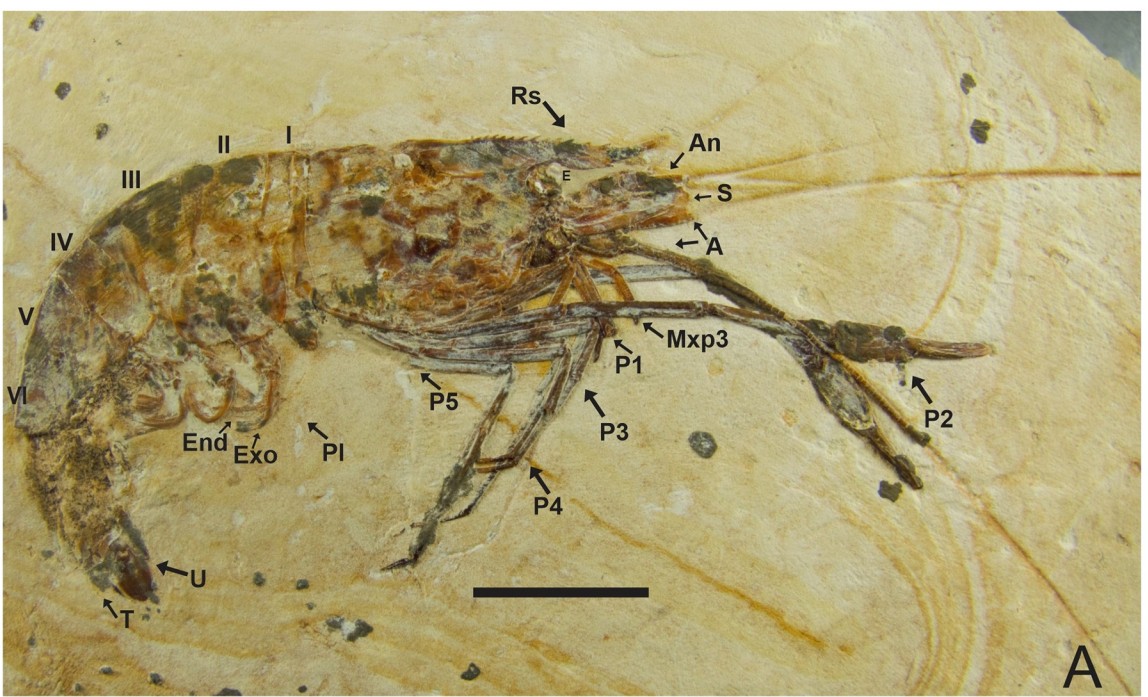

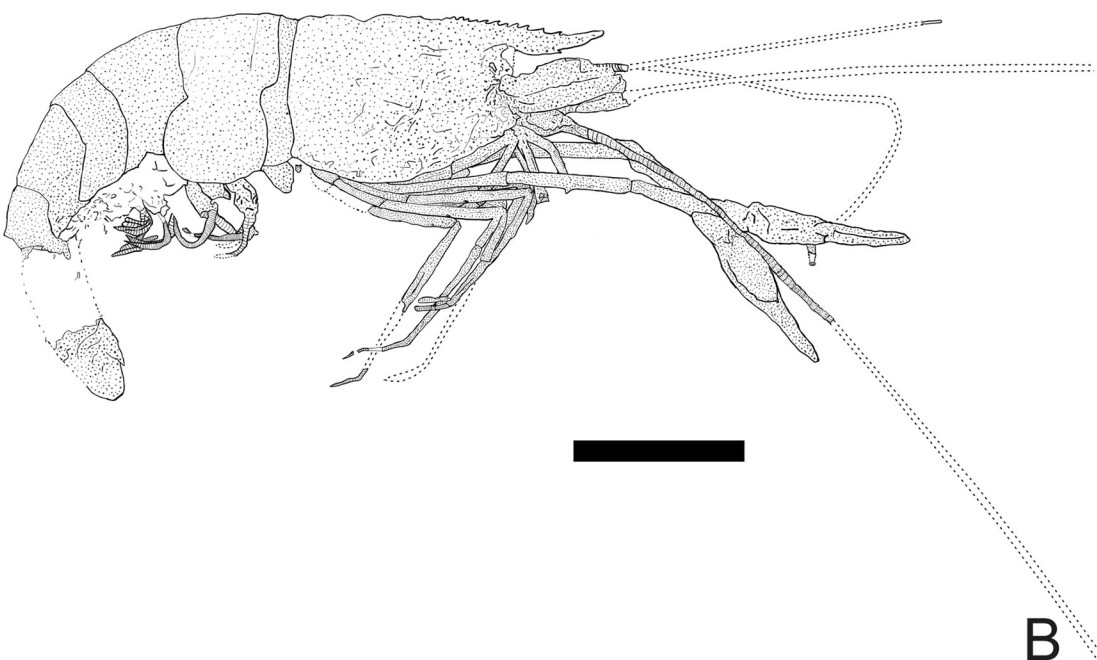

**Fig 5. Lateral view of the specimen 1673/LPU.** (A) Detail of the second pereiopod is represented by chelipeds. (B) Reconstitution, dotted lines represent appendages not preserved but with impressions on the matrix. Rs: Rostrum; An: Antennule; A: Antenna; S: Scaphocerite; Mxp3: Third maxilliped; P1-P5: Pereiopods; Pl: Pleopods; End: Endopod; Exo: Exopod; I-VI: Pleon; T: Telson; U: Uropods. Scale bars: 10 mm. Photo and drawing by Olga A. Barros.

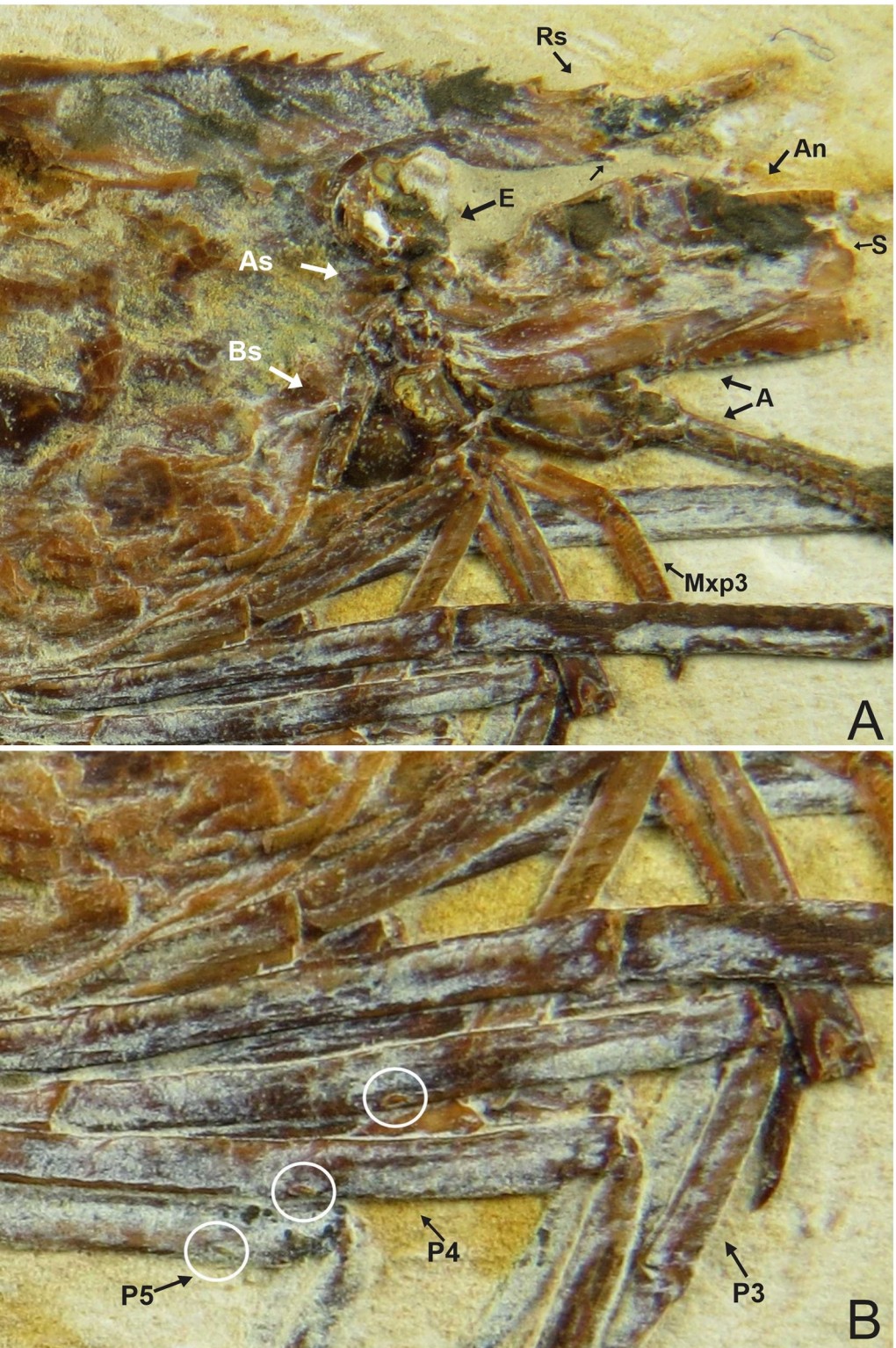

**Fig 6. 1673/LPU.** (A) The rostrum exhibits 12 supra-rostral spines with serrate appearance, two sub-rostral spines clear visible; brush's presence of hairs (bristles) and spines well evident in the third maxilliped. (B) Detail of the Pereiopods P3-P5 have one spine in the merus. E: Eyes; Rs: Rostrum; An: Antennule; A: Antenna; S: Scaphocerite; As: Antennal spines; Bs: Branchiostegal spines; Mxp3: Third maxilliped. Photo by Olga A. Barros.

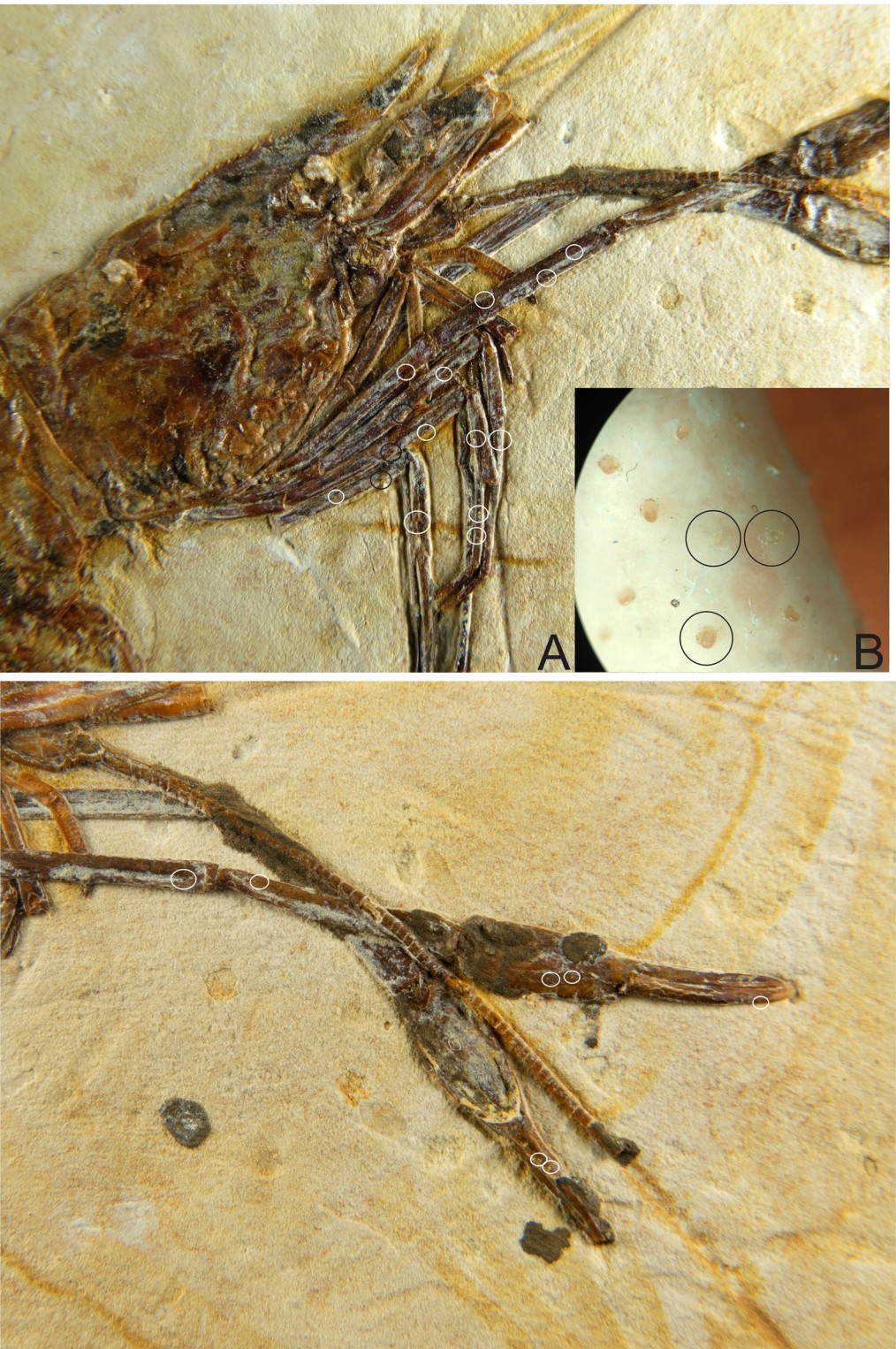

**Fig 7. 1673/LPU.** (A) Detail of the pereiopods spines (black circle) and tubercles (white circle). (B) Detail of tubercles in existing caridean, when the tubercle is broken stay one hole in place of the tubercles (hole in detail of two black circles on the side by side). This hole can be filled by sediment during fossilization. (C) white circle is the detail of the tubercles in the second pereiopod (P2).

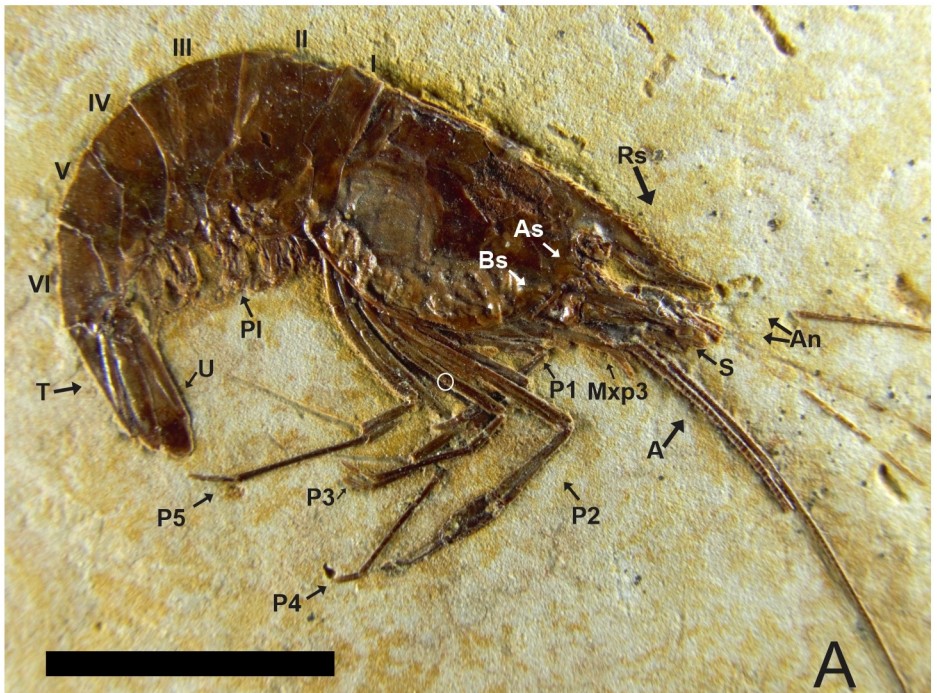

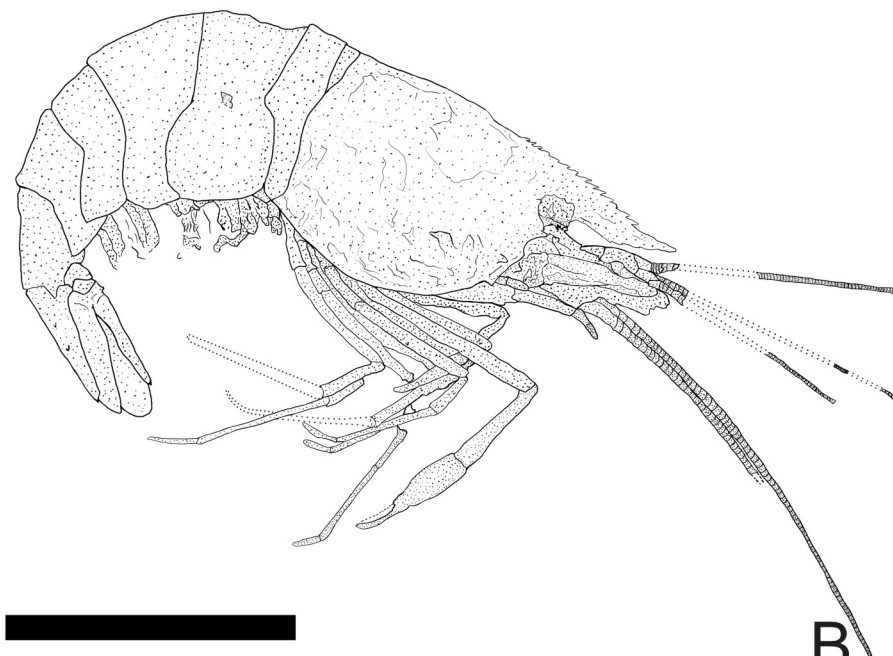

**Fig 8. 2108/MPSC (URCA).** (A) Lateral view of the 2108/MPSC (URCA). (B) Reconstitution, dotted lines represent appendages not preserved but with impressions on the matrix. Rs: Rostrum; An: Antennule; A: Antenna; As: Antennal spines; Bs: Branchiostegal spines; S: Scaphocerite; Mxp3: Third maxilliped; P1-P5: Pereiopods; Pl: Pleopods; I-VI: Pleon; T: Telson; U: Uropods. Scale bars: 10 mm. Photo and drawing by Olga A. Barros.

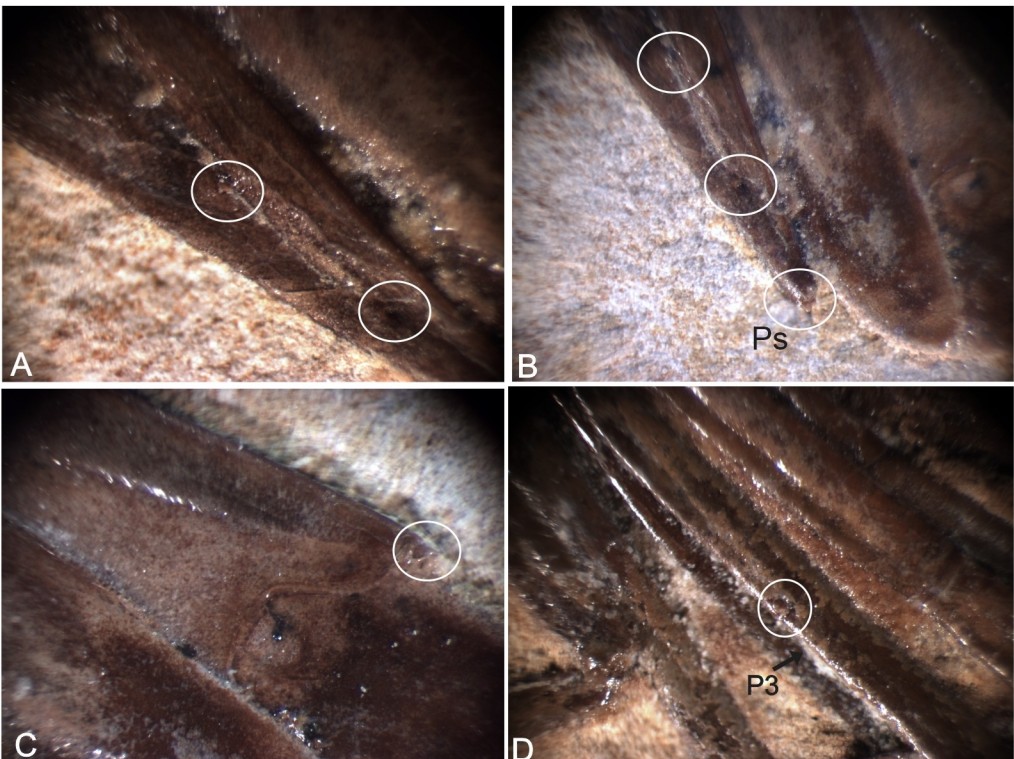

**Fig 9. 2108/MPSC (URCA).** (A) Detail of the telson with one pair of spines. (B) Detail of the telson with the posterior spine (PS). (C) Detail of the uropod. Exopodite with diaresis has crossed by a thin longitudinal carina running parallel to the outside lateral margin and ending with a small spine in highlighted. (D) third pereiopod (P3) with an evident spine.

slightly longer than the telson; protopodite subrectangular in outline and supports the expopodite. The exopodite, with rounded diaresis, is crossed by a thin longitudinal carina, extending parallel to the outside lateral margin and ending with a small spine. Endopodite without ornamentation.

## Description of 171/CR (UFRJ) (Figs 10–12)

**Description.** Medium-sized specimen with well-preserved exoskeleton; total length approximately 25 mm; antennae and antennule very long, but total length is unknown because of the fragmentary nature of specimen; laminar scaphocerite preserved; rostrum partially preserved, with 7 well evident supra-rostral spines, the fragmented end of the rostral tip seems to have 9 supra-rostral spines with serrate appearance, 2 sub-rostral spines clearly visible; laminar scaphocerite present; smooth pleon without spines, pleura of the second somite strongly rounded, overlapping the first and third. Fourth and fifth somites bear a strongly acute pleura; first pereopod (P1) is not discernible, second pereopod (P2) enlarged and chelate and faintly preserved; tubercles on merus, carpus, and propodus; pereiopods (P3-P5) long and narrow; pleopods visible but poorly preserved; only small proximal fragments of telson and uropod are preserved.

## Description of 243/CR (UFRJ) (Figs 13–16)

**Description.** This sample is approximately 28 mm long; antennae and antennules very long, but their total lengths are unknown because of the fragmentary nature of the specimen;

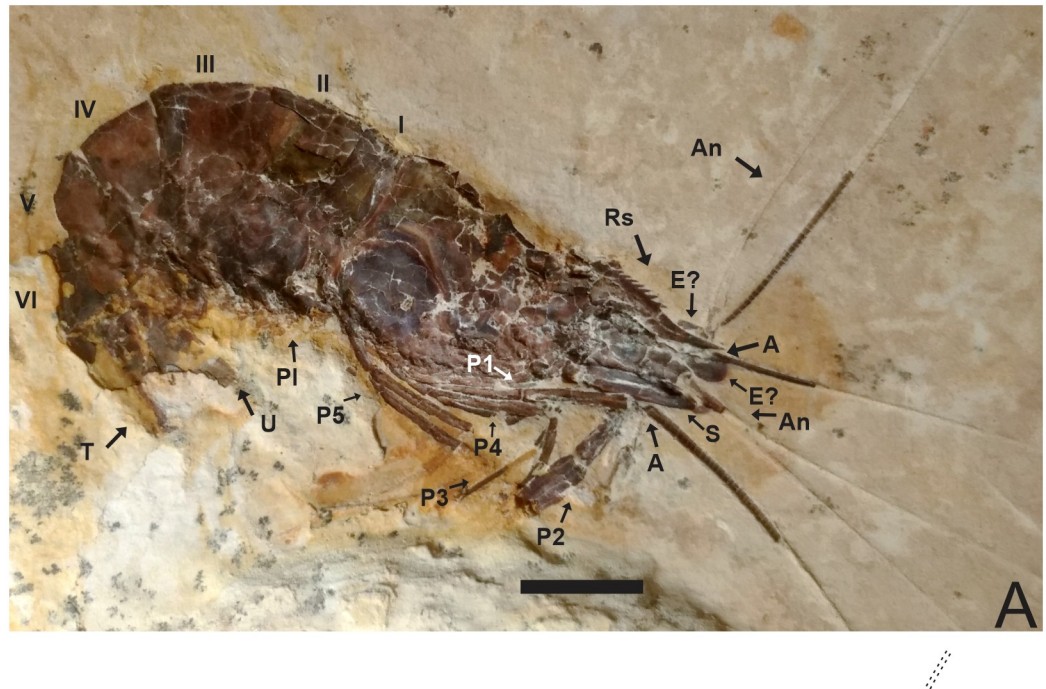

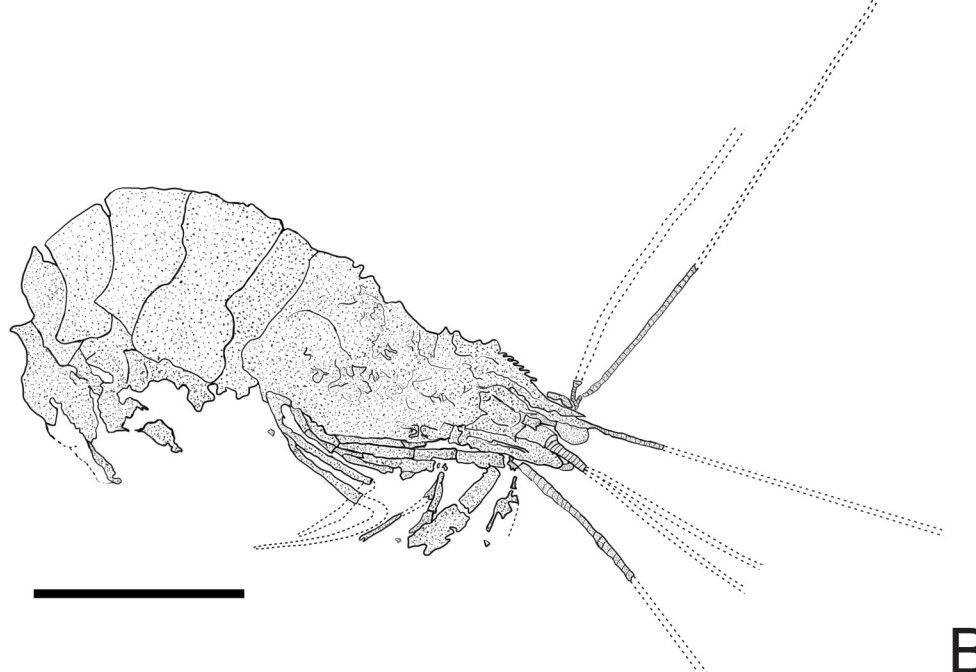

**Fig 10. Lateral view of the UFRJ 171/CR.** (A) second pereiopod are represented by cheliped. (B) Reconstitution, dotted lines represent appendages not preserved but with impressions on the matrix. Rs: Rostrum; An: Antennule; E: Eyes; A: Antenna; S: Scaphocerite; P1-P5: Pereiopods; Pl: Pleopods; I-VI: Pleon; T: Telson; U: Uropods. Scale bars: 10 mm. Photo and drawing by Olga A. Barros.

rostrum partially preserved, with 9 well evident supra-rostral spines and the fragmented end of the rostral tip seems to have 11 rostral spines with serrate appearance; sub-rostral spine not discernible; laminar scaphocerite present; antennal and branchiostegal spines not discernible in carapace; smooth pleon without spines, pleura of second somite strongly rounded,

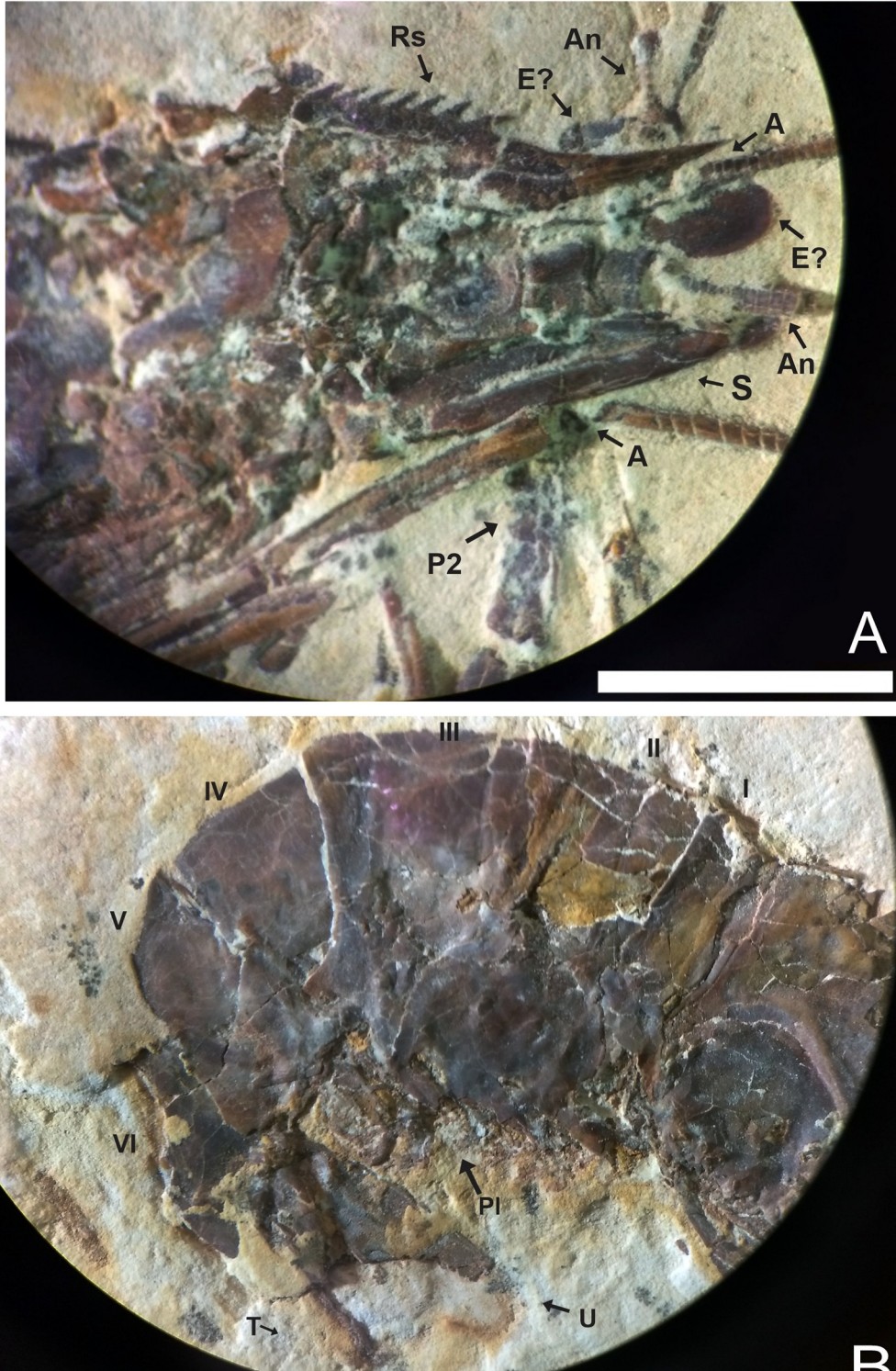

**Fig 11. UFRJ 171/CR.** A) The detail of the rostrum partially preserved show 9 supra-rostral spines with serrate appearance, and 2 sub-rostral spines clearly visible. B) Smooth pleon without spines, the pleura of the second somite is strongly rounded, overlapping the first and third. The fourth and fifth somites are with a strongly acute pleura, characteristic that typical of *Beurlenia araripensis*. Rs: Rostrum; An: Antennule; E: Eyes; A: Antenna; S: Scaphocerite; P2: Second pereopod; I-VI: Pleon; Pl: Pleopods; T: Telson; U: Uropods. Scale bar 5 mm. Photo: Olga A. Barros.

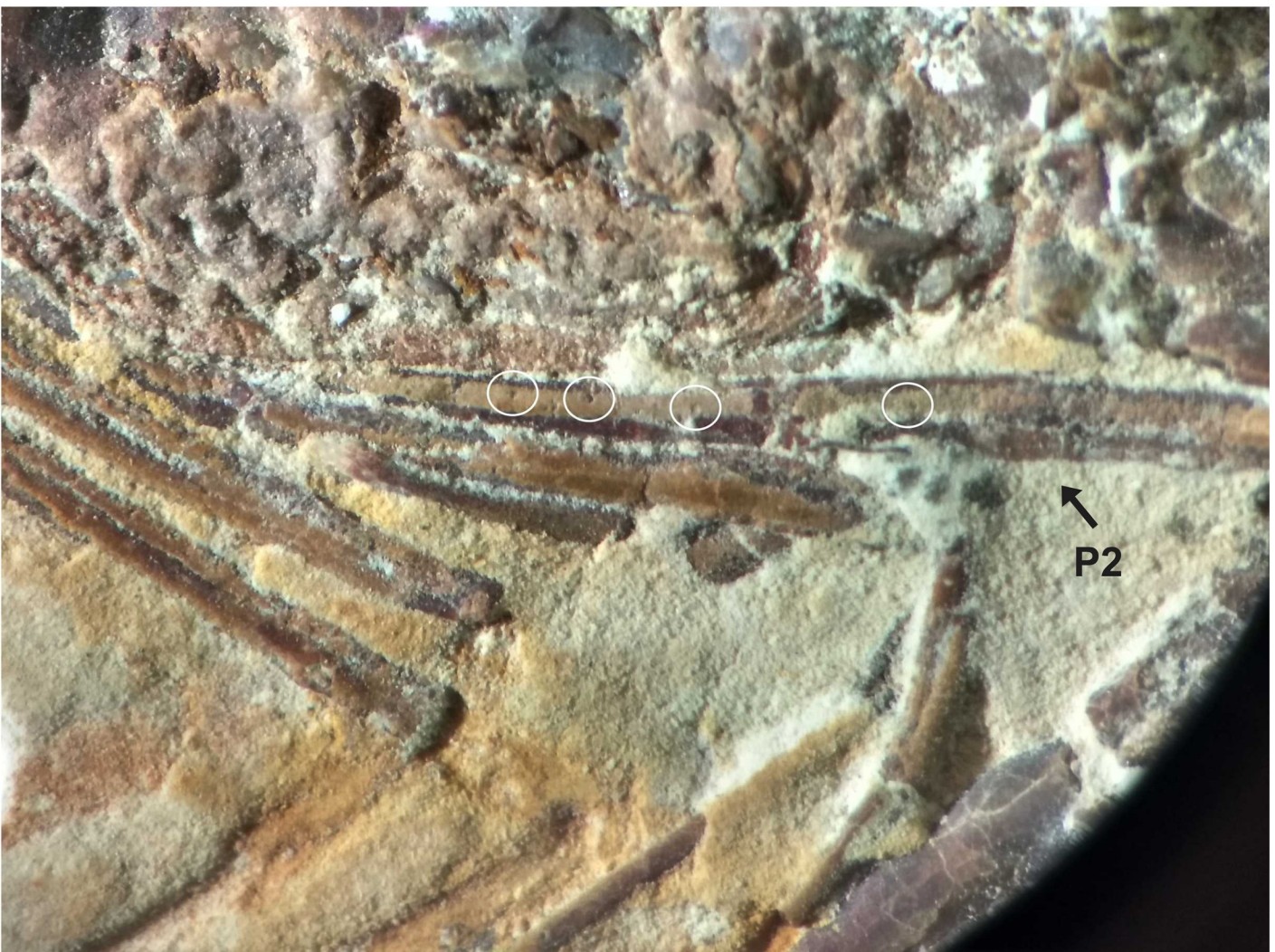

**Fig 12. UFRJ 171/CR.** Detail of the second pereiopods (P2) represented by chelipeds with several tubercles clearly in the merus and carpus. Photo by Olga A. Barros.

overlapping first and third. Fourth and fifth somites with a strongly acute pleura. First pereiopod (P1) incomplete, second pereiopod (P2) enlarged and chelate with elongated propodus, three subsequent pairs of pereiopods (P3-P5) have several tubercles on merus and carpus and are long and narrow; pleopods are partially visible; telson and uropod preserved, but posterior spines not discernible.

## Discussion

The assignment of these samples to Caridea is clear. The second somite with sub-rounded pleura partly overlapping those of somites I and III and the second pereiopod represented by chelipeds are diagnostic for the group.

We easily distinguish the samples analyzed herein from another Caridea found in the Santana group, *Kellnerius jamacaruensis*. Our samples have smooth pleon without spines or groove, whereas the third pleonal somite of *Kellnerius jamacaruensis* has a distinct groove in the first third of the tergite, not extending to the pleurite. The fourth and fifth somites have a

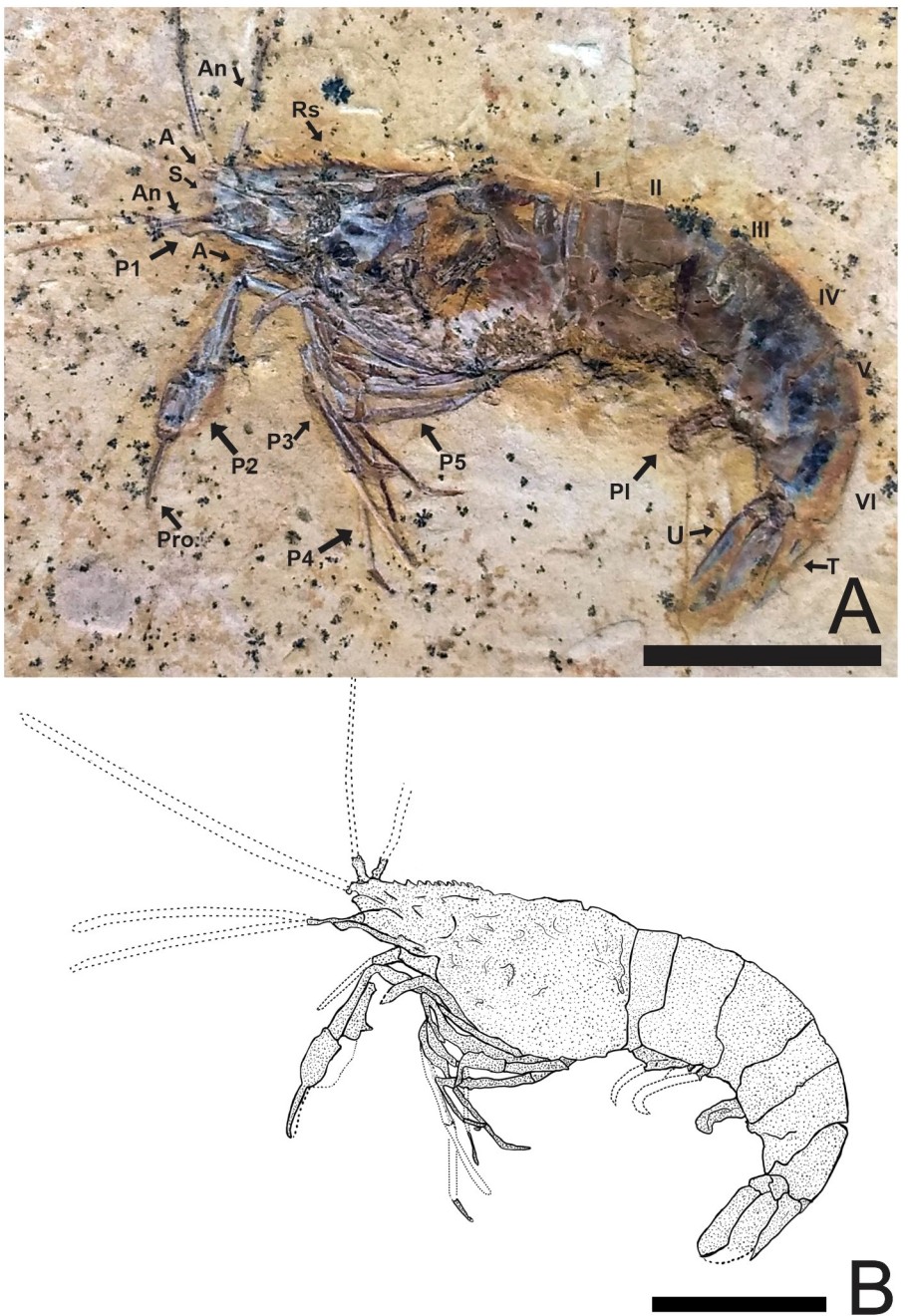

**Fig 13. C UFRJ 243/CR.** (A) second pereiopod are represented by cheliped. B) Reconstitution, dotted lines represent appendages not preserved but with impressions on the matrix. Rs: Rostrum; A: Antenna; An: Antennule; S: Scaphocerite; I-VI: Pleon; P1-P5: Pereiopods; Pro: Propodus; Pl: Pleopods; T: Telson; U: Uropods. Scale bars: 5 mm. Photo and drawing by Olga A. Barros.

strongly acute pleura, versus second and fourth somites about the same size, third and fifth larger (third is the largest), and sixth elongated, in *Kellnerius jamacaruensis*.

Caridean shrimps are very rare in the fossil record and their morphological features are not easy to recognize due to their often poor state of preservation [18]. Accordingly, Martins-Neto *et al.* [4] could not clearly observe the number of rostral spines in *B. araripensis*, but Maisey

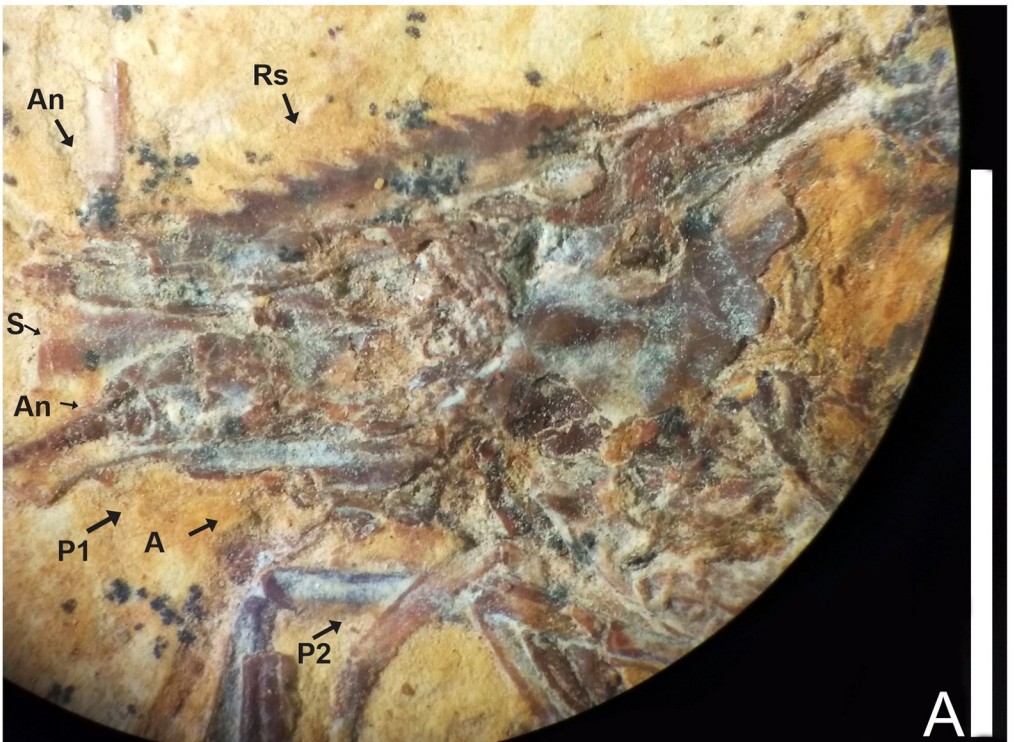

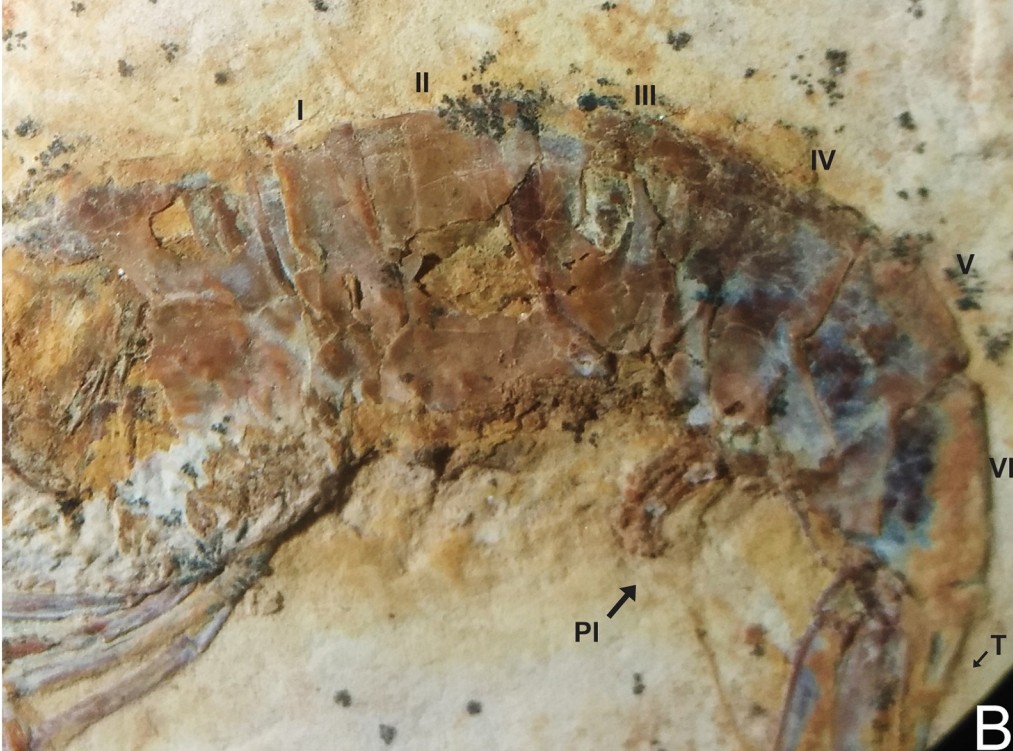

**Fig 14. UFRJ 243/CR.** A) Detail of the rostrum with exhibits 11 supra-rostral spines with serrate appearance, sub-rostral spines not discernible. Rs = rostrum, A = antenna, An = antennule, S = scaphocerite, P1, P2 = pereiopods. B) The pleura of the second somite is strongly rounded, overlapping the first and third. The fourth and fifth somites are with a strongly acute pleura, characteristic that typical of *Beurlenia araripensis*. Rs: Rostrum; A: Antenna; An: Antennule; S: Scaphocerite; P1, P2: Pereiopods; I-VI: Pleon; Pl: Pleopods; T: Telson. Scale bar: 10 mm. Photo by Olga A. Barros.

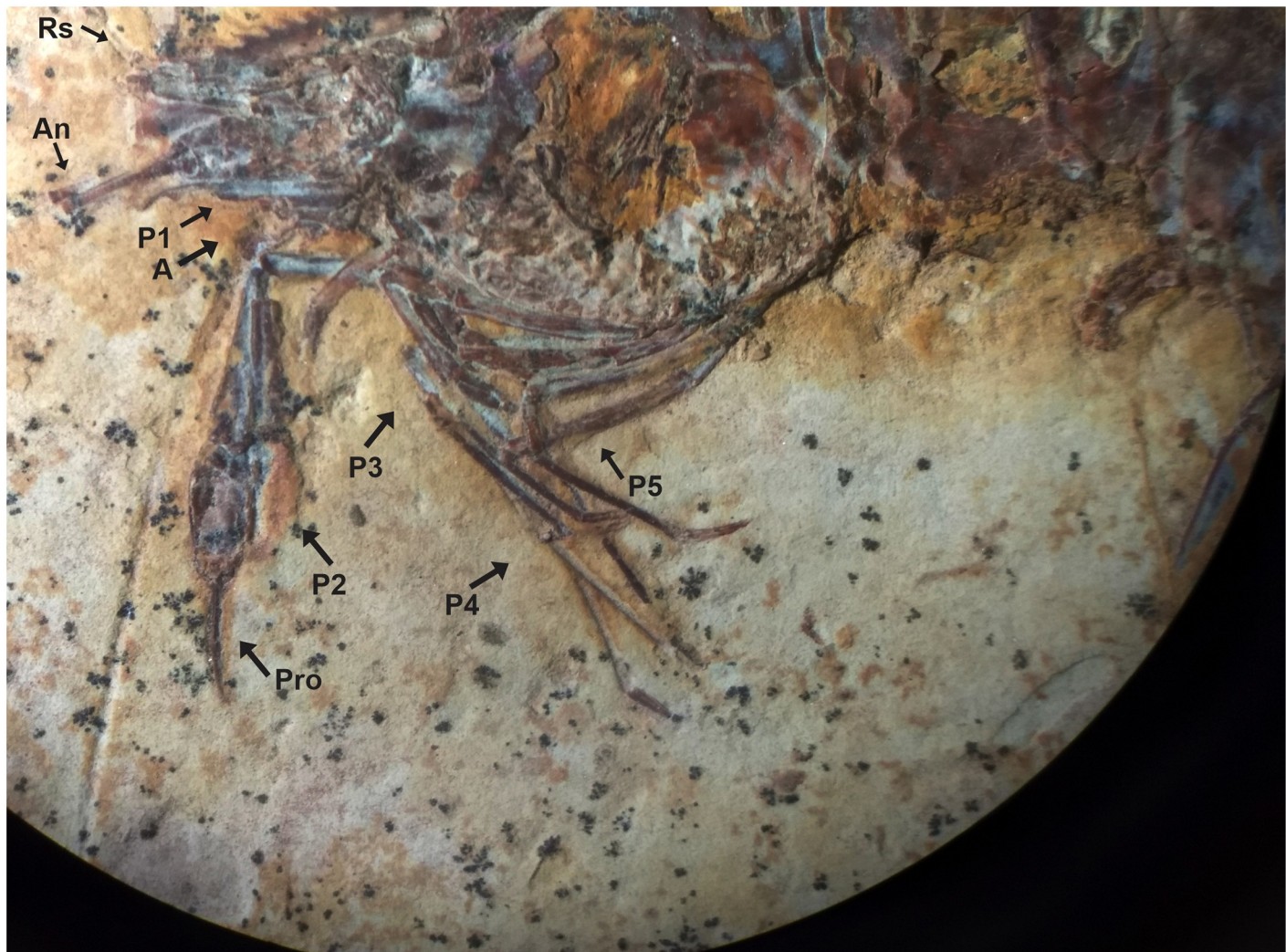

**Fig 15. Lateral view of the UFRJ 243/CR.** Rs: Rostrum; A: Antenna; An: Antennule; S: Scaphocerite; P1-P5: Pereiopods; Pro: Propodus. Photo by Olga A. Barros.

*et al.* [5] recognized 12 small supra-rostral spines in a single analyzed specimen with complete rostrum. Decades later, 14 supra-rostral spines and three sub-rostral spines were recognized [7]. This number of rostral spines is essential for the identification of caridean species [19], and this variation probably reveal morphologic plasticity [7] as occurs, for example, in living species of *Macrobrachium* Bate, 1868 [20,21] and *Palaemon* Weber, 1795 [22], within Palaemonidae Rafinesque, 1815.

The assignment of *B. araripensis* to Palaemonidae, as suggested by Martins-Neto *et al* [4], was confirmed in this work based on the antennal and branchiostegal spines observed on the carapace, at least one pair of articulated spines on the distal extremity of the telson and one articulated spine in the posterior extremity of telson [7]. Other Palaemonidae Rafinesque, 1815, traits of the samples analyzed in this work include: carapace with well-developed antennal and branchiostegal spines (2466/CRT, 1673/LPU, 2108/MPSC); exopodite in the third maxilliped with articulated endopodite (2108/MPSC); first two pairs of pereiopods chelate, with chelae of the second generally larger than those of the first (all samples); last three pairs of

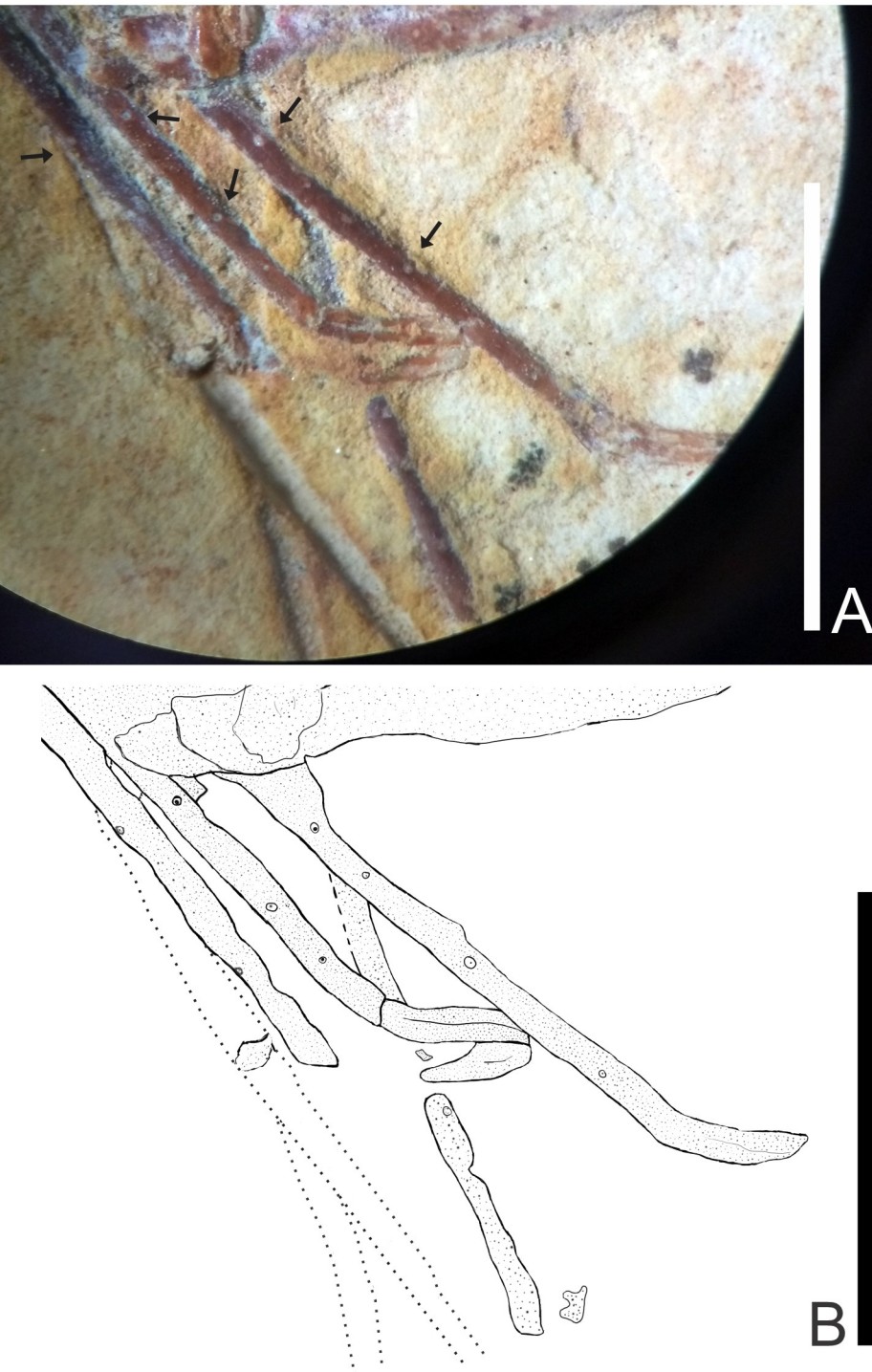

**Fig 16. UFRJ 243/CR.** A) Pereiopods 3–5 have several tubercles in the merus and carpus. B) Reconstitution, dotted lines represent appendages not preserved but with impressions on the matrix. Scale bars: 5 mm. Photo and drawing by Olga A. Barros.

pereopods with simple dactylus (all samples); spines on the distal extremity of the telson (2466/CRT, 2108/MPSC).

We observed in our samples variation in the number of rostral spines. 2466/CRT (UFC) has 5 supra-rostral spines and 3 sub-rostral spines with serrate appearance. 1673/LPU (URCA) has 12 supra-rostral spines with serrate appearance and 2 sub-rostral spines clearly visible. 2108/MPSC (URCA) has 14 supra-rostral and 3 sub-rostral spines. The partially preserved rostrum of 171/CR (UFRJ) has 9 supra-rostral spines with serrate appearance, and 2 sub-rostral spines clearly visible. 243/CR (UFRJ) has a partially preserved rostrum with 11 supra-rostral spines with a serrate appearance, the sub-rostral spines are not clear.

This variation in a five-specimens sample of *B. araripensis* reveals the probable morphologic plasticity of the species, as occurs in living carideans, as *Palaemon ivonicus* Holthus, 1950, *Palaemon carteri* Holthus, 1950, *Palaemon mercedae* Pereira, 1986, and *Palaemon yuna* Carvalho, Magalhães & Mantelatto, 2014. In fact, it is still unclear if the morphological variation found in these species represent only intraspecific variation and may encompass a larger alpha-diversity [23,24].

*Palaemon ivonicus* and *P. carteri* occurs sympatrically in the Amazon basin and are very similar morphologically. These two species have been distinguished primarily based on rostral characters (number of the rostral spines and rostral shape) and on the position of the branchiostegal spine. These characteristics are not enough to consistently differentiate *P. ivonicus* from *P. carteri* [19,23,25]. Therefore, the high interspecific similarity and intraspecific variability found in these species raised doubts whether they represent distinct biological entities [20,22–24]. Molecular data on shrimps, considering the genetic structures of *P. ivonicus* and *P. carteri*, may show the existence of cryptic species. Comparative studies are needed to assess if variations in other taxa of the group are inter- or intraspecific [23].

Observing the complexity in defining species of living shrimps, as opposed to intraspecific variation, one sees the higher complexity of analyzing fossils preserved in two-dimensions. The genus *Palaemon* was previously mentioned as fossils from the Profeti region, Italy, with two species. *Palaemon antonellae* Garassino & Bravi, 2016, exhibits subrectangular carapace; long rostrum with 11 suprarostral teeth protruding forward and six subrostral teeth; somite II with subround pleura partly overlapping those of somite I and III; pereiopods I-II with strong, elongated chelae; telson with two pairs of spines on dorsal surface and one pair of spines on the distal extremity; exopodite with diaeresis [18]. *Palaemon vesolensis* Bravi, Coppa, Garassino & Patricelli, 1999, has a subrectangular carapace, long rostrum with 7 suprarostral teeth forwards protruded and 3 subrostral teeth, somite II with subround pleura partly overlapping that of somite I and III, antennae three-flagellate, Pereiopods I-II chelate with elongate merus and carpus, telson with two pairs of spines on dorsal surface and one pair of spine to the distal extremity, exopodite with diaeresis [26]. As discussed above, the variation of rostrum spines in Palaemonidae is clear, and the differences between *P. antonellae* and *P. vesolensis* basically occurs in the shape and number of rostral spines. Thus, it should be better investigated if we are dealing here with two valid species or with intraspecific variation.

The determination of new species of the same genus is difficult because of the expressive plasticity occurring in Palaemonidae, mainly in the variation of the number of rostral spines. In the case of *Beurlenia araripensis*, preservation aspects bring additional difficulty to the problem. The absence of antennal and branchiostegal spines in the samples described by previous authors [4,5] may be related to the state of preservation of the carapace. In our study, it was not possible to identify the branchiostegal spines in two specimens due to the compaction of the carapace. These new features reported here for *B. araripensis* are important not only for taxonomic, but also for evolutionary and ecological studies.

We believe that the second pereiopod (P2) of *B. araripensis* has several tubercles, but only P3, P4, and P5 have one evident spine. Distal parts of pereiopods have several tubercles, but this structure is well preserved only in some specimens. The spines and tubercles of living caridean are very delicate, but when these structures are broken either by a diagenetic activity or in life, a small hole remains. Such a hole can be covered by the sediment during fossilization and is not clearly observed in some fossils. We believe that 2108/MPSC (URCA) has one spine in pereiopods P3-P5, but this spine is clear only in the third element (P3). This specimen has a thicker layer of protective resin (Paraloid B72) that makes it difficult to identify the pereiopod spines and tubercles. However, we believe that pereiopods P3-P5 have spines and tubercles like those of 1673/LPU (URCA).

As for the comparative study of sexual dimorphism in extant carideans, Nogueira *et al.* [27] concluded that males of *Macrobrachium brasiliense* Heller, 1862, grow differently from females, and become the largest individuals in the population. The adult individuals invest more energy for the development of the second pair of chelipeds and the second abdominal pleura, indicating that the second pair of chelipeds is used by males in reproductive-related behaviors and the second abdominal pleura is an important part of the incubator chamber of females [28,29]. 1673 LPU (URCA) has chelate second pair of pereiopod and its length is large when compared to that of others specimens of *B. araripensis*. This feature may suggest that 1673 LPU (URCA) is an adult male, but the masculine appendix is not preserved to reinforce this hypothesis.

## Conclusion

Caridean fossils are rare, and their anatomical details are not well known due to a typically poor state of preservation. The information presented here may be valuable in understanding its classification. The number of rostral teeth is of great importance in the discrimination of different genera and species of Caridea, but other characters may also be significant for the identification of *B. araripensis*. For instance, the morphological plasticity with this species is meaningful to understand the evolutionary and environmental stages of these decapods.

## Acknowledgments

The authors are grateful to A.A.F. Saraiva and A.P. Pinheiro (URCA) for the support and for borrowing the material under their care, to W. Santana (USC) for suggestions about this research, and to M. Mendes for the access to the samples from the paleontology laboratory (UFC) and for the laboratory infrastructure. We are indebted to museologist F. Figueiredo and to I.S. Carvalho for access to the collection of the Geological Department (UFRJ).

All necessary permits were obtained for the described study, which complied with all relevant regulations. All relevant data are within the manuscript. No authors have competing interests. The studied fossil specimens are properly housed at the following public institutions: Laboratório de Paleontologia of Universidade Regional do Cariri, Crato, Brazil, Museu de Paleontologia Plácido Cidade Nuvens, Santana do Cariri, Brazil, Laboratório de Paleontologia da Universidade Federal do Ceará, Fortaleza, Ceará, Brazil, and Departamento de Geologia da Universidade Federal do Rio de Janeiro, Rio de Janeiro, Brazil.

## Author Contributions

**Conceptualization:** Olga Alcântara Barros, Maria Somália Sales Viana, João Hermínio da Silva.

**Formal analysis:** Alexandre Rocha Paschoal, Paulo Victor de Oliveira.

**Funding acquisition:** Alexandre Rocha Paschoal, Paulo Victor de Oliveira.

**Investigation:** Olga Alcântara Barros, Maria Somália Sales Viana, Paulo Victor de Oliveira.

**Methodology:** Olga Alcântara Barros, Bartolomeu Cruz Viana.

**Supervision:** Olga Alcântara Barros, Maria Somália Sales Viana, João Hermínio da Silva, Alexandre Rocha Paschoal.

**Visualization:** Maria Somália Sales Viana, Paulo Victor de Oliveira.

**Writing – original draft:** Olga Alcântara Barros, Maria Somália Sales Viana.

**Writing – review & editing:** Olga Alcântara Barros, Bartolomeu Cruz Viana, Alexandre Rocha Paschoal, Paulo Victor de Oliveira.

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
