## [Decision Letter · Decision Letter 0]

21 Jul 2020

PONE-D-20-20348

New data about Beurlenia araripensis, lacustrine shrimp from Crato Formation, lower Cretaceous of the Araripe Basin, northeastern Brazil, and their morphological variations based on the quantity rostral spine and morphological variation of pereiopods

PLOS ONE

Dear Dr. Barros,

Thank you for submitting your manuscript to PLOS ONE. After careful consideration, we invite you to submit a revised version of the manuscript that addresses the points raised during the review process.

Dear Dr. Barros,

We have secured three revisions of your MS.

Two of the reviewers are generally happy with the MS, one is asking for more substancial revisions.

At this point, I ask you to consider all modifications they suggest (accept or reply to them accordingly) and proceed so submit a revised version of the text.

Thanks for choosing PLOS ONE as the venue to publish your research!

Kind regards,

Max Cardoso Langer

Academic Editor

PLOS ONE

We look forward to receiving your revised manuscript.

Kind regards,

Max Cardoso Langer

Academic Editor

PLOS ONE

Journal Requirements:

2. We note that Figure 1 in your submission contains map images which may be copyrighted.

We require you to either (a) present written permission from the copyright holder to publish these figure specifically under the CC BY 4.0 license, or (b) remove the figure from your submission:

b. If you are unable to obtain permission from the original copyright holder to publish these figure under the CC BY 4.0 license or if the copyright holder’s requirements are incompatible with the CC BY 4.0 license, please either i) remove the figure or ii) supply a replacement figure that complies with the CC BY 4.0 license. Please check copyright information on all replacement figures and update the figure caption with source information. If applicable, please specify in the figure caption text when a figure is similar but not identical to the original image and is therefore for illustrative purposes only.

4. Your ethics statement must appear in the Methods section of your manuscript. If your ethics statement is written in any section besides the Methods, please move it to the Methods section and delete it from any other section. Please also ensure that your ethics statement is included in your manuscript, as the ethics section of your online submission will not be published alongside your manuscript.

Reviewers' comments:

Reviewer's Responses to Questions

**Comments to the Author**

1. Is the manuscript technically sound, and do the data support the conclusions?

Reviewer #1: Yes

Reviewer #2: Partly

Reviewer #3: Yes

2. Has the statistical analysis been performed appropriately and rigorously? 

Reviewer #1: Yes

Reviewer #2: N/A

Reviewer #3: Yes

3. Have the authors made all data underlying the findings in their manuscript fully available?

Reviewer #1: Yes

Reviewer #2: Yes

Reviewer #3: Yes

4. Is the manuscript presented in an intelligible fashion and written in standard English?

Reviewer #1: No

Reviewer #2: No

Reviewer #3: No

5. Review Comments to the Author

Reviewer #1: please check my notes of the text. The authors must be check carefully my notes in order to improve the text and they must check carefully the used terminology. Finally, they must report author and year for all cited genera and species

Reviewer #2: Comments on Barros et al., new data about Beurlenia

The material is well-preserved and interesting. The paper needs a couple of major issues addressed. One is the numbering and interpretation of chelate pereiopods. The manuscript sometimes suggests that both P1 and P2 are chelate, and other times suggests that is it only P1 or only P2. Remember that the pereiopods come in pairs; it looks to me for example, like there is one pair of strongly chelate pereiopods, probably P2, and the other pair is not well preserved (P1).

The topic of sexual dimorphism is interesting and needs to be expanded. The Italian specimens are mentioned, but they are not designated as to male or female. The differences between males and females in general are not mentioned. This should be discussed, and in addition, the features of the new specimens should also be discussed specifically in terms of sexual dimorphism.

The English grammar needs to be corrected. I marked many corrections.

1. There are three titles for this paper, two on the original PLOS ONE submission page and one on the top of the manuscript. All three are grammatically incorrect. One needs to be picked and made correct.

2. In the reviewer attachment, I made many grammatical corrections and marked sentences that I did not understand/could not correct.

3. Tertiary is no longer used; please use Paleogene or Cenozoic.

4. Under geologic setting, please introduce the Santana Group first, then address the constituent formations.

5. Lines 118-119: I don’t know what the phrase “it pairs the rostrum…” means.

6. Please use pleon instead of abdomen. Abdomen is now regarded as a part of the body that does not bear appendages, whereas a pleon bears appendages.

7. Line 123: the first two pereiopods—do you mean the first 2, or the first 2 pairs? This issue occurs elsewhere also.

8. Lines 133-134: can you be more clear about the lengths? I am not clear on what these legnths refer to (carapace, carapace + pleon, etc.).

9. Line 139: states that the carapace is smooth, yet lines 119 and 120 mention antennal and branchiostegal spines. Can this be reconciled?

10. Line 142: ocular incision is narrow and unidentified. Please clarify. If it is unidentified, how do we know it is narrow?

11. Line 144: first and second maxilla preserved. This would be truly extraordinary, and it needs to be illustrated.

12. Line 146: four subsequent pereiopods…. In carideans, pereiopods 1 and 2 are chelate and pereiopod 3-5 are achelate. This manuscript bounces back and forth on which pereiopods are chelate and which are not. This needs to be clarified. This occurs in lines 166-169 also.

13. Line 259: smaller than carpal. What is smaller than the carpus?

14. Paragraph beginning on line 272: Are you saying that P. antonellae is one sex and P. vesolensis is another? Lines 303-304 suggest that sexual dimorphism exists, but it is not explained. The possible dimorphisms need to be discussed, then the Italian species can be introduced. Then, this needs to be applied to the Beurlenia under consideration here.

15. A note about gender vs sex: In English now, gender refers to the social role or personal identification of a person whereas sex refers to biological features (see Wikipedia). So sex is the preferred term, not gender.

16. Branchiostegal: this is a ventral carapace feature, but usage herein suggests dorsal (“superior” in the text). Please check.

Fig 3, line 454: the other pereiopods—do you mean 3-5?

Fig. 5: telson spines cannot be seen.

Fig 16: first two pereiopods: do you mean both P1 or P1 and P2?

Fig. 18. The last three pairs, not first three.

These are nice specimens and this will make a nice contribution, once these issues are clarified.

Signed,

Carrie E. Schweitzer

Reviewer #3: Most of the comments are on the attached pdf. The big problem is that the manuscript was not edited by a native English-speaking individual. This problem was addressed on the manuscript. The content of the article is fine and the illustrations are very good,except for the shadowing on the line drawings.

6. PLOS authors have the option to publish the peer review history of their article (what does this mean?). If published, this will include your full peer review and any attached files.

Reviewer #1: **Yes: **Alessandro Garassino

Reviewer #2: **Yes: **Carrie E Schweitzer

Reviewer #3: **Yes: **Rodney Feldmann

---

## [Author Response · Author response to Decision Letter 0]

19 Nov 2020

Reviewers' comments: We note that Figure 1 in your submission contains map images which may be copyrighted. All PLOS content is published under the Creative Commons Attribution License (CC BY 4.0), which means that the manuscript, images, and Supporting Information files will be freely available online, and any third party is permitted to access, download, copy, distribute, and use these materials in any way, even commercially, with proper attribution. For these reasons, we cannot publish previously copyrighted maps or satellite images created using proprietary data, such as Google software (Google Maps, Street View, and Earth). For more information, see our copyright guidelines: http://journals.plos.org/plosone/s/licenses-and-copyright. We require you to either (a) present written permission from the copyright holder to publish these figures specifically under the CC BY 4.0 license, or (b) remove the figure from your submission: a. You may seek permission from the original copyright holder of Figure 1 to publish the content specifically under the CC BY 4.0 license. We recommend that you contact the original copyright holder with the Content Permission Form (http://journals.plos.org/plosone/s/file?id=7c09/content-permission-form.pdf) and the following text: “I request permission for the open-access journal PLOS ONE to publish XXX under the Creative Commons Attribution License (CCAL) CC BY 4.0 (http://creativecommons.org/licenses/by/4.0/). Please be aware that this license allows unrestricted use and distribution, even commercially, by third parties. Please reply and provide explicit written permission to publish XXX under a CC BY license and complete the attached form.” Please upload the completed Content Permission Form or other proof of granted permissions as an "Other" file with your submission. In the figure caption of the copyrighted figure, please include the following text: “Reprinted from [ref] under a CC BY license, with permission from [name of publisher], original copyright [original copyright year].”

Answers: The location map of the Araripe Basin (Figure 1) was produced the program QGIS Geographic Information System (version 3.12 - QGIS.org). Open Source Geospatial Foundation Project. 'http://qgis.org” and utilized with based the coordinate system geographic (Datum, SIRGAS 2000), database cartographic IBGE 2017 and CPRM. The stratigraphy of the Santana group was made by the authors with program Corel draw 2020. All images have no copyright and are created by the authors themselves.

Reviewer #1: Please check my notes of the text. The authors must be check carefully my notes in order to improve the text and they must check carefully the used terminology. Finally, they must report author and year for all cited genera and species

Answers: The referee is right. It was substituted the number of the text (reference) for author and, year for all cited genera and species.

Reviewer #2: Comments on Barros et al., new data about Beurlenia. The material is well-preserved and interesting. The paper needs a couple of major issues addressed. One is the numbering and interpretation of chelate pereiopods. The manuscript sometimes suggests that both P1 and P2 are chelate, and other times suggests that is it only P1 or only P2. Remember that the pereiopods come in pairs; it looks to me for example, like there is one pair of strongly chelate pereiopods, probably P2, and the other pair is not well preserved (P1).

Answers: The referee is right. The preservation of P1 is not clear in some samples. In the specimens mentioned by Maisey & Carvalho 1995, and Martins Neto & Mezzalira 1991 the authors do not mention P1 clearly because of the poor preservation, in Saraiva et al. 2019 the authors mention that they are chelated P1 and P2. We decided to include one more specimen in the discussion of the paper (the same specimen analyzed by Saraiva et al. 2019 - MPSC 2108, and we made new images) because the first pereiopod is better preserved in this sample. When we calmly observe this specimen, we observe that there are preserved structures in the fossil that were not mentioned by Saraiva et al. 2019. For example, one spine in P3 (like in our samples) and the preservation of of the third maxilliped. We made a proper photo of telson spines, we mentioned this in the text and we made new images of the sample MPSC 2108.

Reviewer #2: The topic of sexual dimorphism is interesting and needs to be expanded. The Italian specimens are mentioned, but they are not designated as to male or female. The differences between males and females in general are not mentioned. This should be discussed, and in addition, the features of the new specimens should also be discussed specifically in terms of sexual dimorphism.

Answers: The different Italian species are mentioned in the text to demonstrate that the two Palaemonidae can be very similar. The word “gender” was used by mistake in the beginning of the paragraph of the original text of the manuscript, that sentence was replaced by “The genus Palaemon was previously mentioned in crustacean fossils from the Profeti region, Italy, with two species, the first was Palaemon antonellae Garassino & Bravi, 2016, which…”. This way, we aim to avoid any ambiguity in our text. Palaemon antonellae and Palaemon vesolensis were differentiated basically by the number of spines of the rostrum and the second pereopod is much larger in Palaemon antonellae. We compared these specimens with our samples because we believe that we do not have more than one species of Beurlenia being differentiated only by the number of rostral spines. We believe that there is an intraspecific variation observed in the number of rostral spines as occurs in current Palaemonidae; Macrobrachium Bate 1868 and Palaemon Weber 1795. In summary, we did not intend to discuss the topic of sexual dimorphism related to the Italian specimens and the text was modified to make it clearer.

Reviewer #2: The English grammar needs to be corrected. I marked many corrections.

Answers: The reviewer is correct. The English grammar was carefully reviewed and corrections were made.

Reviewer #2: There are three titles for this paper, two on the original PLOS ONE submission page and one on the top of the manuscript. All three are grammatically incorrect. One needs to be picked and made correct. 

Answers: The reviewer is correct. The title it was corrected the English grammar and was replaced.

Reviewer #2: 2. In the reviewer attachment, I made many grammatical corrections and marked sentences that I did not understand/could not correct.

Answers: The referee is right, English corrections were made.

Reviewer #2: 3. Tertiary is no longer used; please use Paleogene or Cenozoic.

Answers: The reviewer is correct. It was substituted for Paleogene.

Reviewer #2: 4. Under geologic setting, please introduce the Santana Group first, then address the constituent formations.

Answers: The Santana group was included in the text in line 81 "According to Assine [14], the Araripe Basin is subdivided stratigraphically into pre-rift and post-rift (I and II) sequences formed mainly by fluvial and lacustrine strata, where crops out a rich fossiliferous deposit known as the Santana Group. This group comprising the Rio da Batateira, Crato, Ipubi, Romualdo, and Arajara formations (Aptian-Albian in age)”.

Reviewer #2: 5. Lines 118-119: I don’t know what the phrase “it pairs the rostrum…” means.

Answers: The reviewer is correct. The English grammar was corrected and was substituted for “rostrum with laminar scaphocerite”.

Reviewer #2: 6. Please use pleon instead of abdomen. Abdomen is now regarded as a part of the body that does not bear appendages, whereas a pleon bears appendages.

Answers: The referee is right. It was substituted for “pleon”.

Reviewer #2: 7. Line 123: the first two pereiopods—do you mean the first 2, or the first 2 pairs? This issue occurs elsewhere also.

Answers: The referee is right. First 2 pairs - P1 and P2 are chelates.

Reviewer #2: 8. Lines 133-134: can you be more clear about the lengths? I am not clear on what these legnths refer to (carapace, carapace + pleon, etc.).

Answers: The referee is right. Was replaced to total length 46 mm.

Reviewer #2: 9. Line 139: states that the carapace is smooth, yet lines 119 and 120 mention antennal and branchiostegal spines. Can this be reconciled?

Answers: Only the pleon is smooth, and in the carapace of some specimens, it is possible to see clearly the antennal and branchiostegal spines. These parts are not clearly seen in other specimens because of the poor preservation states of the fossils.

Reviewer #2: 10. Line 142: ocular incision is narrow and unidentified. Please clarify. If it is unidentified, how do we know it is narrow?

Answers: The reviewer is correct, it has been removed from the text.

Reviewer #2: 11. Line 144: first and second maxilla preserved. This would be truly extraordinary, and it needs to be illustrated.

Answers: After a second detailed analysis we see that these structures are not preserved in the fossil (there was confusion in the interpretation of pereiopods, not is first and second maxilla).

Reviewer #2: 12. Line 146: four subsequent pereiopods…. In carideans, pereiopods 1 and 2 are chelate and pereiopod 3-5 are achelate. This manuscript bounces back and forth on which pereiopods are chelate and which are not. This needs to be clarified. This occurs in lines 166-169 also.

Answers: The referee is right; the correction was made.

Reviewer #2: 13. Line 259: smaller than carpal. What is smaller than the carpus?

Answers: The referee is right; it has been removed from the text and a new paragraph was made.

Reviewer #2: 14. Paragraph beginning on line 272: Are you saying that P. antonellae is one sex and P. vesolensis is another? Lines 303-304 suggest that sexual dimorphism exists, but it is not explained. The possible dimorphisms need to be discussed, then the Italian species can be introduced. Then, this needs to be applied to the Beurlenia under consideration here.

Answers: P. antonellae and P. vesolensis are two different species that differ only in the number of rostral spines. Palaemon antonellae have the second pereiopods (chela) much larger than Palaemon vesolensis. In some species of current Palaemonidae, there is sexual dimorphism observed by the size of the second pereiopods (the chela is greater in adult males). In our specimens of Beurlenia, we believe that there is variation in the number of the rostral spines, like it occurs in existing species, and that the size of the chela is related to the male adult individual, in opposite to the two species mentioned above.

Reviewer #2: 15. A note about gender vs sex: In English now, gender refers to the social role or personal identification of a person whereas sex refers to biological features (see Wikipedia). So sex is the preferred term, not gender.

Answers: The referee is right. Was substituted to sex.

Reviewer #2: 16. Branchiostegal: this is a ventral carapace feature, but usage herein suggests dorsal (“superior” in the text). Please check.

Answers: The referee is right, it has been corrected in the text.

Reviewer #2: Fig 3, line 454: the other pereiopods—do you mean 3-5?

Answers: Yes, it was added in the text.

Reviewer #2: Fig. 5: telson spines cannot be seen.

Answers: Arrows were placed on the images and another specimen in which the telson spines are clearly observed was added. In addition, the fossil was wetted so that we are now able to view more clearly the tubercles, spines of the pereiopods and telson spines.

Reviewer #2: Fig 16: first two pereiopods: do you mean both P1 or P1 and P2?

Answers: P1 and P2

Reviewer #2: Fig. 18. The last three pairs, not first three.

Answers: The reviewer is correct, in the sentence, the correct is “The last three pairs” (P3-P5).

These are nice specimens and this will make a nice contribution, once these issues are clarified.

Signed,

Carrie E. Schweitzer

Reviewer #3: Most of the comments are on the attached pdf. The big problem is that the manuscript was not edited by a native English-speaking individual. This problem was addressed on the manuscript. The content of the article is fine and the illustrations are very good, except for the shadowing on the line drawings.

Answers: The manuscript was fully reviewed so that English language mistakes were corrected. Line drawings on the illustrations were edited.

---

## [Editor Report · Decision Letter 1]

18 Jan 2021

PONE-D-20-20348R1

New data about Beurlenia araripensis, lacustrine shrimp from Crato Formation, Lower Cretaceous of the Araripe Basin, northeastern Brazil, and his morphological variations based on the shape and the number of rostral spines

PLOS ONE

Dear Dr. Barros,

Thank you for submitting your manuscript to PLOS ONE. After careful consideration, we feel that it has merit but does not fully meet PLOS ONE’s publication criteria as it currently stands. Therefore, we invite you to submit a revised version of the manuscript that addresses the points raised during the review process.

All the requested modification are include in the attached file PONE-D-20-20348R1 editor track.docx.

We look forward to receiving your revised manuscript.

Kind regards,

Max Cardoso Langer

Academic Editor

PLOS ONE

Additional Editor Comments (if provided):

Dear authors,

We are happy to accept your MS for publication, as long as you can modify it according to the editor's comment attached as a MSWord file with track changes (PONE-D-20-20348R1 editor track.docx).

Sincerely,

Max Langer

---

## [Author Response · Author response to Decision Letter 1]

21 Jan 2021

We thank the Reviewers for the considerations that have made a great contribution to the improvement of our work.

All grammatical changes in the text were made, as well as the exclusions of phrases suggested by the reviewer.

Reviewers' comments: Use open nomenclature, i.e. no taxonomic categories

Answers: The suggested replacement was made.

Reviewers' comments: Include a reference to the stratigraphy here.

Answers: It was addition to the text the reference Valença et al. [9]

Reviewers' comments: Slender than what?

Answers: It has been substituted for “Exopodite is smaller than ischium and merus”

(1) Thank you for updating your data availability statement. You note that your data are available within the Supporting Information files, but no such files have been included with your submission. At this time we ask that you please upload your minimal data set as a Supporting Information file, or to a public repository such as Figshare or Dryad.

Please also ensure that when you upload your file you include separate captions for your supplementary files at the end of your manuscript.

As soon as you confirm the location of the data underlying your findings, we will be able to proceed with the review of your submission.

We have uploaded the data in Figshare: 0.6084 / m9.figshare.13623023 (https://figshare.com/articles/figure/New_data_on_Beurlenia_araripensis_Martins-Neto_Mezzalira_1991/13623023)

(2) We note that your submission includes two sets of figures, in the set uploaded on July 1, 2020, there is Figure 17 and Figure 18 which are not referred to in the current version of the manuscript. Please remove duplications and Please ensure that you refer to all figures in your text as, if accepted, production will need this reference to link the reader to the figure.

Thanks, we have removed the old and duplicated figures.

---

## [Editor Report · Decision Letter 2]

9 Feb 2021

New data on Beurlenia araripensis Martins-Neto & Mezzalira, 1991, a lacustrine shrimp from Crato Formation, and its morphological variations based on the shape and the number of rostral spines

PONE-D-20-20348R2

Dear Dr. Barros,

We’re pleased to inform you that your manuscript has been judged scientifically suitable for publication and will be formally accepted for publication once it meets all outstanding technical requirements.

Kind regards,

Max Cardoso Langer

Academic Editor

PLOS ONE

---

## [Editor Report · Acceptance letter]

18 Feb 2021

PONE-D-20-20348R2 

New data on *Beurlenia araripensis* Martins-Neto & Mezzalira, 1991, a lacustrine shrimp from Crato Formation, and its morphological variations based on the shape and the number of rostral spines. 

Dear Dr. Barros:

I'm pleased to inform you that your manuscript has been deemed suitable for publication in PLOS ONE. Congratulations! Your manuscript is now with our production department. 

Kind regards, 

on behalf of

Dr. Max Cardoso Langer 

Academic Editor

PLOS ONE